# X-ray Total Scattering Study of Phases Formed from Cement Phases Carbonation

Ana Cuesta *, Angeles G. De la Torre and Miguel A. G. Aranda

Departamento de Química Inorgánica, Cristalografía y Mineralogía, Universidad de Málaga, 29071 Málaga, Spain; mgd@uma.es (A.G.D.l.T.); g_aranda@uma.es (M.A.G.A.)
* Correspondence: a_cuesta@uma.es

**Abstract:** Carbonation in cement binders has to be thoroughly understood because it affects phase assemblage, binder microstructure and durability performance of concretes. This is still not the case as the reaction products can be crystalline, nanocrystalline and amorphous. The characterisation of the last two types of components are quite challenging. Here, carbonation reactions have been studied in alite-, belite- and ye'elimite-containing pastes, in controlled conditions (3% $CO_2$ and RH = 65%). Pair distribution function (PDF) jointly with Rietveld and thermal analyses have been applied to prove that ettringite decomposed to yield crystalline aragonite, bassanite and nano-gibbsite without any formation of amorphous calcium carbonate. The particle size of gibbsite under these conditions was found to be larger (~5 nm) than that coming from the direct hydration of ye'elimite with anhydrite (~3 nm). Moreover, the carbonation of mixtures of C-S-H gel and portlandite, from alite and belite hydration, led to the formation of the three crystalline $CaCO_3$ polymorphs (calcite, aragonite and vaterite), amorphous silica gel and amorphous calcium carbonate. In addition to their PDF profiles, the thermal analyses traces are thoroughly analysed and discussed.

**Keywords:** building materials; amorphous phases; nanocrystalline phases; carbonation; Rietveld quantitative phase analysis; pair distribution function; thermal analysis

## 1. Introduction

Concrete, made from Portland cement, water, aggregates, mineral additives and admixtures, is the second largest commodity (in terms of total volume) consumed by mankind after water [1]. The yearly concrete world production is estimated to be 5 tons/person or ~7 km³ (~18 Gt) [2] and the estimated world concrete stock is 315 Gt [3] which currently results in ~4 Gt/yr of concrete demolition waste (CDW) [4,5]. Therefore, any chemical reaction in concretes is of huge societal interest because durability of current and future buildings and infrastructures is key to maintaining our well-being.

Carbon dioxide from our atmosphere may react with hydrated/hydrating cement phases in presence of moisture. The carbonation reactions can have a profound impact on: (i) phase assemblage, (ii) binder microstructure and (iii) the durability performances of (reinforced) concretes. A great research effort in carbonation of building materials has been done and this investigation focusses on the phase changes and not on the impact of carbonation on building materials performance properties. The interested reader is addressed to current reviews for overviews of the different processes and their consequences in pastes, mortars and concretes [6–8]. In this work, the research is focussed on the phase changes under carbonation and hence, microstructure and durability modifications are not further considered, although it is acknowledged that these transformations are equally important. Furthermore, carbonation-provoked changes can be considered passive or active. On the one hand, passive carbonation processes cause unwanted and unplanned changes in the outer layer of a concrete exposed to the environment and it is commonly considered as a deterioration (or ageing) mechanism. On the other hand, an active carbonation process results from an intentionally designed procedure to take advantage of the ability

of calcium-containing cement phases to react with $CO_2$ resulting in a variety of benefits including improved early age mechanical, chemical or environmental performances. Here, just passive changes are considered.

Table 1 summarises the carbonation reactions of the main components in hydrated cement binders which are further investigated here. The current knowledge, related to the phases involved in these reactions, is briefly summarised next. I. Portlandite, crystalline calcium hydroxide, carbonates according to reaction (1) to yield calcium carbonate(s). The crystalline structure(s) of the $CaCO_3$ form will mainly depend upon the dominant conditions during precipitation which could be kinetic or thermodynamic [7]. $CaCO_3$ will precipitate as aragonite, vaterite or even partly containing amorphous calcium carbonate (ACC) under kinetically controlled conditions. Conversely, under a thermodynamically controlled environment, calcite will mainly crystallize. II. Carbonation of C-S-H gel, see reaction (2), proceeds by the removal of calcium ions yielding, initially through a de-calcified gel plus partial carbonation, and finally the formation of amorphous (calcium-containing) silica gel and several forms of $CaCO_3$ [9–14]. The extent of the C-S-H carbonation and the type of product formed depend upon the initial binder (Ca/Si ratio of the gel, water content, overall phase assemblage, etc.) as well as the environmental conditions (mainly: time of exposure, $CO_2$ concentration and the relative humidity (RH)). All calcium carbonate polymorphs and even ACC have been reported as a result of reaction (2). It has been also published that aragonite and vaterite formation seem to be related to extensive C-S-H carbonation likely due to prolonged times, to high $CO_2$ concentration or a mixture of both. III. Ettringite carbonates according to reaction (3) to yield calcium carbonate(s), calcium sulphate(s) and nanocrystalline gibbsite although alumina gel has also been reported to form [15–18]. Calcium carbonate polymorph formation seems to depend upon the carbonation conditions with works reporting aragonite [16,19], vaterite [20], mixtures of vaterite and aragonite [18], calcite [17] and even the three $CaCO_3$ polymorphs depending upon the investigated sample [15]. It is well established that gypsum is formed at high RH values of ~90% meanwhile bassanite forms at lower RH (e.g., 60–70%) [18,19,21]. Although older research works referenced alumina gel as a result of AFt (ettringite phase) decomposition [20], recent investigations report it as aluminium hydroxide [15–17]. AFm-type phases, monosulfoaluminates, are also carbonate in a similar way than that of AFt, leading to calcium carbonates, calcium sulphates and nanocrystalline gibbsite/aluminium hydroxide [18]. Finally, the authors are not aware of the works fully devoted to prove the stability of aluminium hydroxide under $CO_2$ atmospheres, see reaction (4). However, from the reports of carbonation of calcium aluminate and sulphoaluminate cement binders [15–17], it could be deduced that $Al(OH)_3$ is not carbonated, which is backed by other publications [22].

**Table 1.** Carbonation reactions of selected components in hydrated cement binders [7].

| Component | Chemical Formula | Carbonation Reaction (in $H_2O$ Presence) | Reaction |
|---|---|---|---|
| Portlandite | $Ca(OH)_2$ | $Ca(OH)_2 + CO_2 \rightarrow CaCO_3$ [#] $+ H_2O$ | (1) |
| C-S-H-gel * | $(CaO)_xSiO_2 \cdot nH_2O$ * | $(CaO)_{1.8}SiO_2 \cdot 4H_2O + 1.8CO_2 \rightarrow 1.8CaCO_3 + SiO_2 \cdot nH_2O + yH_2O$ | (2) |
| Ettringite, AFt | $Ca_6Al_2(SO_4)_3(OH)_{12} \cdot 26H_2O$ | $AFt + 3CO_2 \rightarrow 3CaCO_3 + 3CaSO_4 \cdot mH_2O$ [$] $+ 2Al(OH)_3 + zH_2O$ | (3) |
| Gibbsite | $Al(OH)_3$ | $Al(OH)_3 + CO_2 \rightarrow$ stable? | (4) |

[#] $CaCO_3$ can precipitate as calcite, aragonite, vaterite, amorphous calcium carbonate(s) or mixtures of these phases. * Calcium silicate hydrate gel composition is variable but close to x~1.7–1.9 and n~4.0 for neat Portland cement pastes [23,24]. [$] Calcium sulphate precipitates as gypsum or bassanite depending upon the RH of the carbonation environment.

As discussed just above, there is an understanding in the precipitation of calcium sulphate forms but there is controversy in the formation of different calcium carbonates. Furthermore, in addition to the three crystalline $CaCO_3$ polymorphs, ACC could also form. It is far from straightforward to disentangle the formation of amorphous phases within these systems, which also contain initial amorphous/nanocrystalline phases like C-S-H gel [25,26] or nano-gibbsite [27–29]. To further study amorphous/nanocrystalline phases, synchrotron total scattering pair distribution function (PDF) methodology [30,31] is an

ideal tool. Collecting a total scattering data with enough Q-resolution, Rietveld quantitative phase analyses could be carried out in the reciprocal-space and PDF analyses in the direct-space. In fact, PDF analyses have been recently carried out for better understanding of the local structure of C-S-H gel which could be satisfactorily explained as a mixture of nanocrystalline defective clinotobermorite and an amorphous component with an atomic local order of few layer-thick calcium hydroxide [23,32,33]. PDF methodology has been recently used for improving our understanding in cement chemistry. In situ synchrotron and neutron PDF were employed to study the hydration of alite and G-type oil well cement including the role of a phosphonate retarder [34]. It was shown that calcium complexation by the retarder was key, with C-S-H gel nuclei poisoning and CH partial precipitation inhibition also contributing. In situ synchrotron PDF was also employed to study the atomic structural deformation of C-S-H under external loading. The consequences of applied loading were measured by: (i) the strain gauge, (ii) the d-spacing shifts (reciprocal space), and the interatomic distance shifts (real space) [35]. For the r < 20 Å range (in real space), where the C-S-H contribution dominates, a 53-year-old alite paste had a much higher overall elastic modulus than a 131-day-old paste, being 18.3 and 8.3 GPa, respectively. PDF can also be used to investigate the alternative sustainable binders which may contain more alkalis [36]. Many sodium-substituted calcium-(alumino-)silicate-hydrate gels were studied and it was shown that the addition of higher levels of alkalis resulted in a systematic reduction of the degree of silicate polymerisation of the gels with clear consequences in their nanoscale ordering.

To the best of the authors knowledge, there are just two papers employing PDF to further study carbonation of cementitious materials. In a first work [37], in situ PDF was employed to investigate the local atomic structural changes occurring during the accelerated carbonation of four types of alkali-activated slag cements. C-A-S-H gel decalcification was shown to proceed in the presence of amorphous calcium carbonate. In addition, it was reported that high MgO content favours the formation of ACC over calcite or vaterite. In a second work from the same authors [13], in situ PDF was also employed to investigate the changes during the accelerated carbonation of synthetic C-S-H gel. Rietveld analyses firmly stablished the precipitation of vaterite (~60 wt%) and calcite (~40 wt%). The analysis of the in situ PDF data showed a continuous calcium decalcification of the C-S-H gel and the final products being an amorphous silica-rich decalcified gel. Finally, it is worth underlining that synchrotron PDF has been used to study the local structures of amorphous and crystalline calcium carbonates from organisms [38,39]. Although this work is not directly related to cement binder carbonation, it reports interesting signatures of crystalline and amorphous calcium carbonates in the PDF patterns that can be used for comparative purposes.

In order to contribute to the understanding of the carbonation reactions in cement binders, three reference pastes have been prepared here: (i) ettringite and nano-gibbsite paste from the hydration of ye'elimite with anhydrite; (ii) C-S-H gel with high amount of portlandite from the hydration of $Ca_3SiO_5$; and (iii) C-S-H gel with low amount of portlandite from the hydration of $Ca_2SiO_4$. A fraction of these pastes were carbonated in controlled conditions. The six resulting samples were analysed by synchrotron total scattering, by Rietveld and pair distribution function approaches. Furthermore, thermal analyses were also carried out. The polymorphism of the different forms has been firmly established from high-resolution synchrotron data. It is proved that ettringite decomposed, in the employed experimental conditions, to yield aragonite, bassanite and nano-gibbsite. It is explicitly stated that ACC in not formed. Conversely, the mixtures of C-S-H gel and portlandite gave the three crystalline $CaCO_3$ polymorphs and ACC. The signatures of ACC in the thermal analysis traces are now clearly shown up.

## 2. Materials and Methods

### 2.1. Total Scattering Synchrotron X-ray Powder Diffraction

Synchrotron X-ray powder diffraction (SXRPD) patterns for the six pastes were collected at the X-ray powder diffraction BL04-MSPD beamline at ALBA synchrotron (Barcelona,

Spain) in Debye-Scherrer (transmission) mode [40]. The wavelength, 0.41318(1) Å, was selected with a double-crystal Si (111) monochromator and determined by using Si640d NIST standard (a = 5.43123 Å). Five patterns were collected for each sample, each lasting ~40 min, and merged in order to improve the signal-to-noise ratio in the recorded very wide angular range, 1 to 130° (2θ) (for the employed wavelength, 0.41318 Å).

### 2.1.1. Rietveld Analysis

Rietveld method using GSAS [41] v.1.8. software was used to perform quantitative phase analysis. A pseudo-Voigt peak shape function with the asymmetry correction included [42,43] was used. The refined overall parameters were: background coefficients, phase scale factors, unit cell parameters, zero-shift error, peak shape parameters and preferred orientation coefficient if needed. The crystal structures used for all phases are given in the Appendix A (Table A1).

### 2.1.2. PDF Analysis

PDFgetX3 software [44] was used to obtain the PDF experimental data with $Q_{max}$ = 21 Å$^{-1}$. PDFGui software [45] was employed to obtain structural and quantitative phase analysis. Crystalline nickel PDF data were used to determine the instrumental parameters: $Q_{damp}$ = 0.036 Å$^{-1}$ and $Q_{broad}$ = 0.061 Å$^{-1}$. Final global optimised parameters were: scale factors, the unit cells and atomic displacement parameters (ADPs). For the gibbsite phase, the atomic positional coordinates and the sp diameter (Spd, equivalent spherical diameter of the nanoparticle) were also optimised. Moreover, the delta2 value, low-r correlated motion peak sharpening factor [46,47], was also refined in the low-r region fits.

### *2.2. Thermal Analysis (TA)*

Differential thermal analysis (DTA) and thermogravimetric analysis (TGA) measurements for the six pastes were performed in a SDT-Q600 analyser from TA instruments (New Castle, DE). The temperature was varied from room temperature (RT) to 1000 °C at a heating rate of 10 °C/min and with a gas flow speed of 100 mL/min. Measurements were made in open platinum crucibles under synthetic air flow (80% $N_2$ and 20% of $O_2$). The derivatives of the weight loss (DTG) have also been calculated and given in the figures.

### *2.3. Particle Size Distribution (PSD) Analysis*

The particle size distribution of the anhydrous samples was measured by laser diffraction (Malvern MasterSizer S, Malvern, UK) provided with a wet chamber. Powders were previously dispersed in isopropanol using an ultrasonic bath.

### *2.4. BET Surface Area Determination*

The specific surface area of the anhydrous samples was measured by multi-point $N_2$ adsorption with a BET (ASAP 2420, Micromeritics, Norcross, GA, USA) instrument.

### *2.5. Samples Description*

Three anhydrous samples were employed in this study: (i) A mixture of stoichiometric orthorhombic ye'elimite $Ca_4[Al_6O_{12}]SO_4$ [48] with 31.0 wt% of anhydrite labelled as Y-A; (ii) stoichiometric triclinic tricalcium silicate, t-$C_3S$, which was purchased from Mineral Research Processing (M.R.PRO); and (iii) monoclinic β-dicalcium silicate, with chemical formula $Ca_2Si_{0.972}Al_{0.028}O_{3.986}?_{0.014}$ where the symbol ? stands for oxygen vacancies, named as β-$C_2S$, which was prepared as detailed in [49]. This β-dicalcium silicate sample also contains 8.4(3) wt% of γ-dicalcium silicate. Anhydrite, for sample Y-A, was prepared by heating bassanite, marketed by BELITH S.P.R.L. (Belgium), at 700 °C for 1 h. Table 2 gives the particle size distribution details, density and specific surface area of the used anhydrous samples.

**Table 2.** Particle size distribution details, density and specific surface data of the employed anhydrous cement samples *.

| Materials | $D_{v,10}$ | $D_{v,50}$ | $D_{v,90}$ | $\rho$ (g/cm$^3$) | BET (m$^2$/g) |
|---|---|---|---|---|---|
| o-$C_4A_3\overline{S}$ | 1.3 | 7.1 | 22.1 | 2.61(1) | 1.1(1) |
| $C\overline{S}$ | 1.5 | 10.1 | 42.6 | 2.93(1) | 2.6(1) |
| t-$C_3S$ | 0.8 | 4.6 | 11.0 | 3.15(1) | 1.9(1) |
| β-$C_2S$ | 1.0 | 6.5 | 35.7 | 3.22(1) | 1.7(1) |

* Cement notation is used here after: C = CaO, S = SiO$_2$, A = Al$_2$O$_3$, $\overline{S}$ = SO$_3$, H = H$_2$O. Therefore, e.g.,: Ca$_3$SiO$_5$ is termed C$_3$S.

### 2.6. Hydration and Accelerated Carbonation

Pastes were prepared by adding deionised water to the solid material to attain water to solid mass ratios (w/s) of 0.85 for Y-A, 0.50 for t-C$_3$S and 0.45 for β-C$_2$S. Y-A paste was manually stirred for 2 min. t-C$_3$S and β-C$_2$S pastes were mechanically stirred at 800 rpm for 90 s twice, with a stopping time of 30 s between them. The as-prepared pastes were cast into hermetically closed polytetrafluoroethylene cylinders and rotated at 16 rpm at 20 $\pm$ 1 °C (see Figure 1). The pastes were demoulded after 3 days for Y-A, 24 h for t-C$_3$S and 28 days for β-C$_2$S and stored in demineralised water at 20 $\pm$ 1 °C for 1.5 months for Y-A, 5 months for t-C$_3$S and 12 months for β-C$_2$S, (see A in Figure 1). The reason for such long hydration times was to have very large degree of hydration for the three pastes. Then, the cylinders were taken out of the water and the surface was gently dried with paper. To prepare the reference samples, the cylinders were ground by hand in an agate mortar, inserted as powder in an Eppendorf and immediately stored in vacuum sealed bags by applying about 0.85 bars within a standard food vacuum sealer unit. The samples were stored in the vacuum-sealed bag up to the day of data collection (see Figure 1). To prepare the carbonated samples, the cylinders were directly stored in vacuum-sealed bags. Prior to the introduction of the samples in the carbonation chamber, the cylinders were ground by hand in an agate mortar. Accelerated carbonation assay was performed for 80 days at 3% of CO$_2$, at 20 °C and a fixed RH of 65% obtained by a saturated salt solution. The six samples are labelled hereafter: Y-A_hyd, Y-A_carb, C$_3$S_hyd, C$_3$S_carb, C$_2$S_hyd and C$_2$S_carb. It is important to point out that the non-carbonated samples were kept in sealed bags for 80 days, while the remaining samples were carbonated.

Samples (reference and carbonated) were kept in Eppendorfs in vacuum-sealed bags until total scattering synchrotron X-ray powder diffraction (SXRPD) data collection. Glass capillaries of 0.70 mm diameter were filled with the powders and sealed with grease to avoid any alteration of the samples.

Finally, in order to discuss the compositions and the weight losses of the hydrated pastes, the hydration reactions must be stated. Hence, the expected product contents, including their ratios, can be calculated. Table 3 reports the most accepted hydration reactions for ye'elimite, alite and belite, which are reactions (5) to (7), respectively. It should be noted that gibbsite and C-S-H are nanocrystalline gels and therefore, their compositions are approximated.

**Table 3.** Stoichiometries of the hydration reactions for the phases employed in this study [23,33,50].

| Hydration Reactions, With the Amount of Products in [g/mol] | | | Reaction |
|---|---|---|---|
| Ca$_4$[Al$_6$O$_{12}$]SO$_4$ + 2CaSO$_4$ + 38·H$_2$O | → | Ca$_6$Al$_2$(SO$_4$)$_3$(OH)$_{12}$·26H$_2$O [1255.1] + 4 Al(OH)$_3$ [312.0] | (5) |
| Ca$_3$SiO$_5$ + 5.2 H$_2$O | → | (CaO)$_{1.8}$SiO$_2$·4H$_2$O [233.1] + 1.2 Ca(OH)$_2$ [88.9] | (6) |
| Ca$_2$SiO$_4$ + 4.2 H$_2$O | → | (CaO)$_{1.8}$SiO$_2$·4H$_2$O [233.1] + 0.2 Ca(OH)$_2$ [14.8] | (7) |

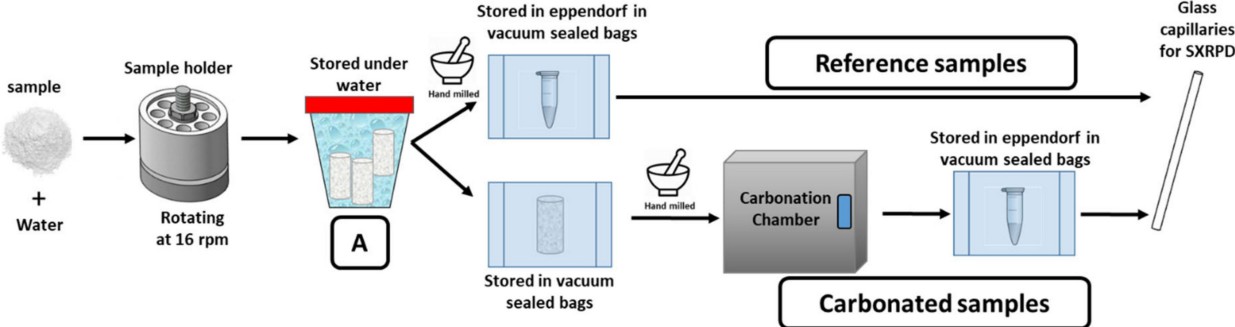

**Figure 1.** Sketch of the experimental procedure employed to prepare the samples.

## 3. Results and Discussion

The total scattering SXRPD raw data for the six studied samples (Y-A_hyd, Y-A_carb, $C_3S$_hyd, $C_3S$_carb, $C_2S$_hyd and $C_2S$_carb,) are plotted in Figure 2. It can be seen that the experimental configuration used allow us to obtain, in addition to a large Q-range for PDF analyses, high resolution (in reciprocal space) data which are suitable for Rietveld analysis. Therefore, in a first stage, the low angle regions of these patterns were analysed by the Rietveld method. In a second stage, the full patterns were inspected and analysed by the PDF method. It is noted here that we have recently discussed the different resolutions arising from common synchrotron total scattering experimental configurations [51].

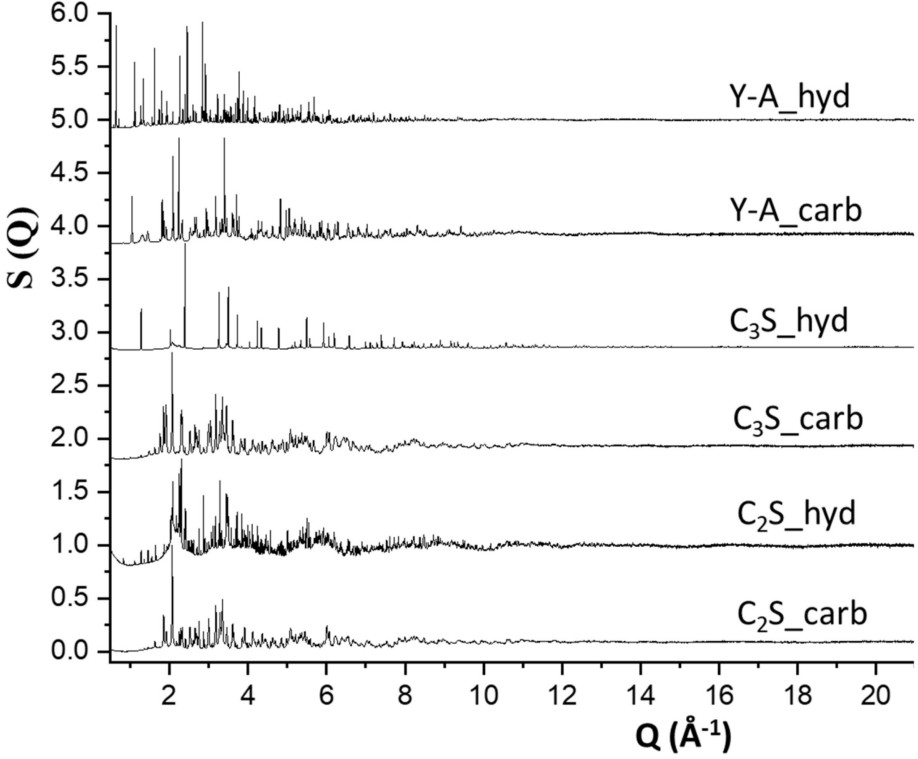

**Figure 2.** Synchrotron X-ray total scattering functions, S(Q), for the studied cement samples. The patterns were normalised with respect to their highest diffraction peaks and vertically displaced for adequate visualisation.

### 3.1. Synchrotron Rietveld Quantitative Phase Analysis

The low angle regions (2–15°/2θ) of the SXRPD patterns were analysed by the Rietveld method as detailed in the experimental section. Figure 3 displays the Rietveld plots for Y-A_hyd and Y-A_carb samples. The Rietveld quantitative phase analysis (RQPA)

results are reported in Table 4. It has been confirmed that the hydration of ye'elimite and anhydrite yields ettringite as the unique crystalline phase and nanocrystalline gibbsite in agreement with previous publications [27,29]. The crystal structure of gibbsite was not used to fit nano-gibbsite diffraction peaks in this RQPA because they were very broad, see Figure 3. Carbonation of this paste resulted in aragonite, as the only $CaCO_3$ crystalline phase. In agreement with previous reports [18,19,21], only bassanite is observed here, as calcium sulphate crystalline phase, because the RH of the carbonation experiment was 65%. Furthermore, nanocrystalline gibbsite displayed broad peaks but relatively sharper than those present in Y-A_hyd. Therefore, the crystal structure of this phase was included in the Rietveld fit and the results are shown in Figure 3 and Table 4.

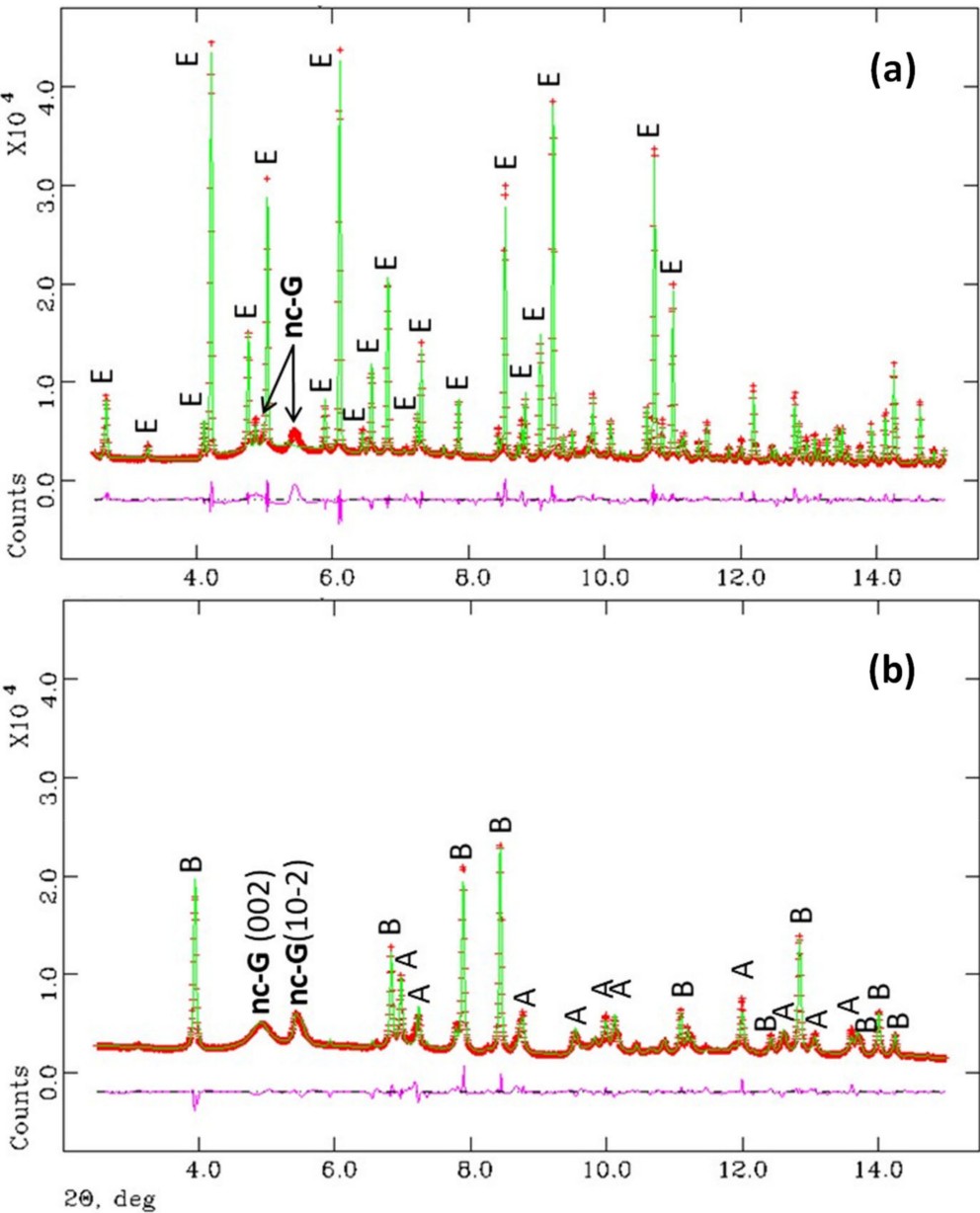

**Figure 3.** Selected low angle ranges (intensity vs 2θ) of the synchrotron Rietveld plots (λ = 0.41 Å) for (**a**) Y-A_hyd, hydrated for 1.5 months and (**b**) Y-A_carb paste after carbonation for 80 days at 3% of $CO_2$ (T = 20 °C and RH = 65%). The top pattern contains crystalline ettringite and very broad diffraction peaks of nano-gibbsite. The main peaks are labelled. Top pattern: ettringite (E) and nanocrystalline-gibbsite (**nc-G**). Bottom pattern: aragonite (A), bassanite (B), nanocrystalline-gibbsite (**nc-G**) [including their (hkl) indexes].

The Rietveld fit for $C_3S\_hyd$ sample is shown in Figure 4. It can be deduced that this is a high-quality sample as the unhydrated $C_3S$ percentage is very low and the sample does not contain crystalline $CaCO_3$ which proves that the sample was properly prepared and stored. The main crystalline phase is portlandite, 93% of the overall crystalline content, as reported in Table 4. However, the main component in this sample is nanocrystalline C-S-H gel being ~67 wt% according to reaction (6) which was not taken into account for the Rietveld quantification due to its nanocrystalline/amorphous nature. When $C_3S\_hyd$ is carbonated, see experimental section, the three crystalline $CaCO_3$ polymorphs are present in the sample in similar amounts. It should also be noted that the maximum in the background, which arises from the contribution of amorphous phase(s), moved from ~8° (~3.05 Å) for $C_3S\_hyd$ to ~7° (~3.5 Å) for $C_3S\_carb$. The presence of this very large broad hump in carbonated pastes was already reported to be characteristic of amorphous silica gel [52].

**Table 4.** Results of the Rietveld quantitative phase analyses of the synchrotron X-ray total scattering patterns.

| Phase (wt%) | Y-A_hyd | Y-A_carb | $C_3S\_hyd$ | $C_3S\_carb$ | $C_2S\_hyd$ | $C_2S\_carb$ |
|---|---|---|---|---|---|---|
| $t\text{-}C_3S$ | - | - | 6.7(1) | - | - | - |
| $\beta\text{-}C_2S$ | - | - | - | - | 55.0(3) | 8.1(1) |
| $\gamma\text{-}C_2S$ | - | - | - | - | 28.7(4) | - |
| CH | - | - | 93.3(4) | 0.6(1) | 8.7(2) | - |
| AFt | 100.0 | - | - | - | - | - |
| Calcite | - | - | - | 26.8(9) | 2.2(2) | 43.1(1) |
| Vaterite | - | - | - | 34.1(3) | - | 5.9(4) |
| Aragonite | - | 23.4(2) | - | 38.5(3) | - | 42.9(1) |
| Bassanite | - | 38.6(1) | - | - | - | - |
| Monocarbonate * | - | - | - | - | 5.4(5) | - |
| nano-Gibbsite | - | 37.9(9) | - | - | - | - |

* Monocarbonate refers to an AFm-type phase with the following chemical formula: $Ca_2Al(OH)_6[(CO_3)_{0.5}\cdot 2.5H_2O]$.

The Rietveld fit for $C_2S\_hyd$ sample was more complicated as shown in Figure 5. After hydration for one year, 55 wt% of $\beta\text{-}C_2S$ and 29 wt% of $\gamma\text{-}C_2S$ remained unhydrated, see Table 4, with respect to 100% of crystalline material. However, it should be noted that the main component in this sample is nanocrystalline C-S-H gel which was not accounted for in this analysis. It must also be highlighted that this sample was partly carbonated with 2.2 wt% of calcite and 5.4 wt% of monocarbonate AFm-type phase likely due to the prolonged hydration time. From this study, several results are obtained. First, $\gamma\text{-}C_2S$ has fully reacted/carbonated. Second, the three crystalline $CaCO_3$ polymorphs are present but the vaterite content is much smaller and the calcite content much higher, when compared to the results obtained for $C_3S\_carb$. Third, the maximum in the background is again moved from ~8° for $C_2S\_hyd$ to ~7° for $C_2S\_carb$, see Figure 5.

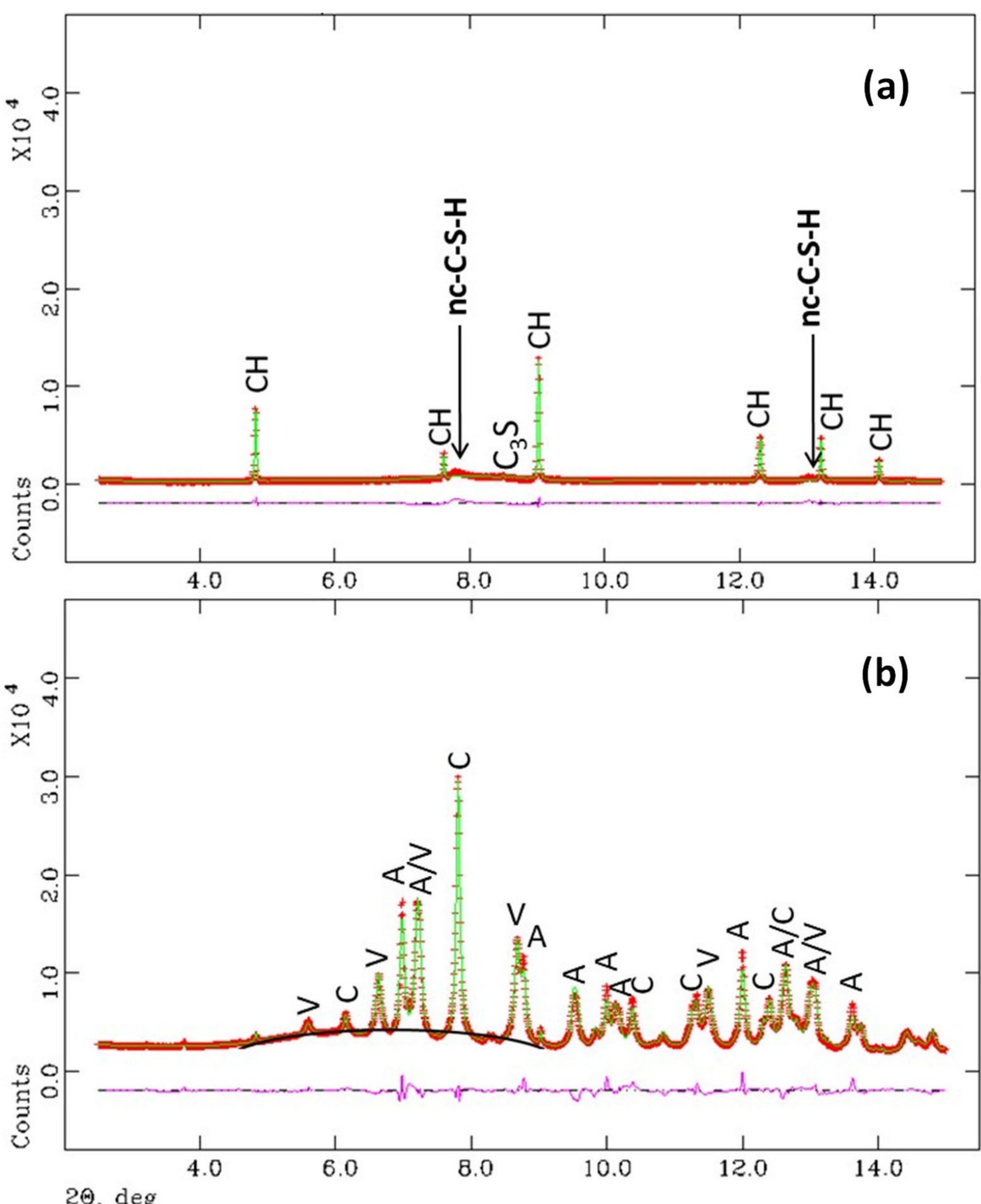

**Figure 4.** Selected low angle ranges (intensity vs. 2θ) of the synchrotron Rietveld plots (λ = 0.41 Å) for (**a**) hydrated t-$C_3S$ for 5 months and (**b**) the $C_3S$ paste after carbonation for 80 days at 3% of $CO_2$ (T = 20 °C and RH = 65%). The main peaks are labelled. Top pattern: Portlandite (CH), alite ($C_3S$) and nanocrystalline C-S-H gel (**nc-C-S-H**). Bottom pattern: calcite (C), aragonite (A) and vaterite (V). The increase in the background centred at ~7°/2θ for $C_3S\_carb$ is highlighted.

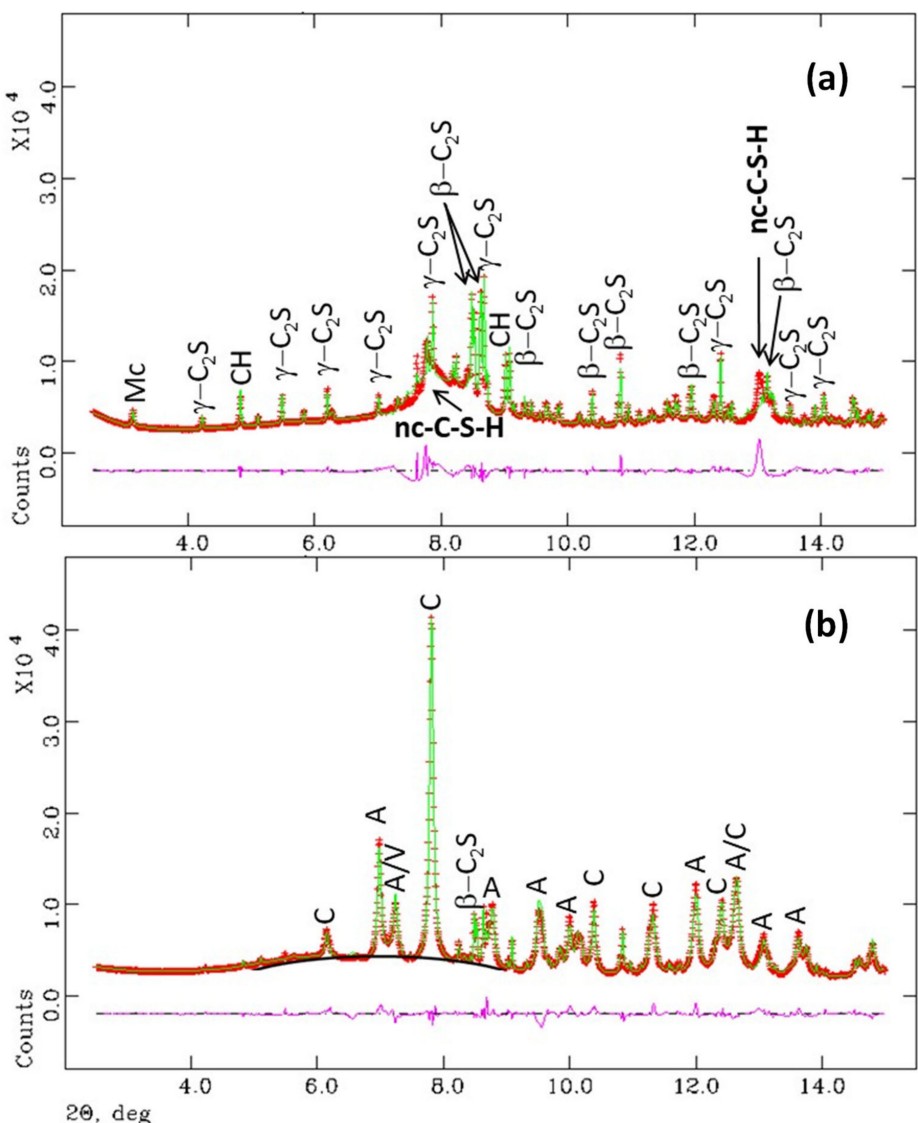

**Figure 5.** Selected low angle ranges (intensity vs. 2θ) of the synchrotron Rietveld plots (λ = 0.41 Å) for (**a**) hydrated β-$C_2S$ for 12 months and (**b**) the $C_2S$ paste after carbonation for 80 days at 3% of $CO_2$ (T = 20 °C and RH = 65%). Top pattern: β-belite (β-$C_2S$), γ-belite (γ-$C_2S$), nanocrystalline C-S-H gel (**nc-C-S-H**), portlandite (CH) and monocarbonate (Mc). Bottom pattern: β-belite (β-$C_2S$), calcite (C), aragonite (A) and vaterite (V). The increase in the background centred at ~7°/2θ for $C_2S$_carb is highlighted.

### 3.2. Synchrotron Pair Distribution Function Analysis

The PDF data were analysed following a multi r-range approach as previously reported [27,29]. The high interatomic correlations region, in this case 4.0–6.0 nm r-range, allows the crystalline phase contents to be determined, meanwhile the low region is used to characterise the atomic ordering in the nanocrystalline/amorphous components. Figure 6a,c displays the high r-range PDF plots for Y-A_hyd and Y-A_carb samples, respectively. The PDF pattern for Y-A_hyd was fitted by using only the crystal structure of ettringite and the final $R_W$ value was 30.4%, which confirmed the Rietveld results showing that this was the only crystalline phase. The analogous PDF analysis for Y-A_carb was very satisfactory by using the crystal structures of bassanite and aragonite, $R_W$ = 23.9%. The last refinement converged to 61 wt% of bassanite and 39 wt% of aragonite. This agrees quite well with the Rietveld results reported in Table 3 after removing the nano-gibbsite contribution being 62.3 and 37.7 wt%, respectively. Furthermore, according to the carbonation

reaction given in Table 1 and without taking into consideration nano-gibbsite, ettringite decomposes to give 59.2 wt% of bassanite and 40.8 wt% of $CaCO_3$. The overall agreement is very good and it shows that (i) the analyses are accurate, and (ii) most of the calcium carbonate is precipitated as crystalline aragonite. Then, the low r-range region, 1.6–4.0 nm, was analysed for these two samples. In this case, the crystal structure of gibbsite was added and its Spd parameter was refined as it takes into account the size of the scattering particles. For Y-A_hyd the refinement converged to Spd = 3.2 nm and the phase contents were 72 and 28 wt% for ettringite and nano-gibbsite, respectively. The Rw value for this fit was relatively low, 26.0%, and a selected region is shown in Figure 6b, the lowest r-range. For Y-A_carb the refinement converged to Spd = 4.6 nm and the phase contents were 34, 21 and 45 wt% for bassanite, aragonite and nano-gibbsite, respectively. The fit is shown in Figure 6d, having $R_W$ = 29.1%. These values are a bit far from the expected results from the chemical reactions (5) and (3): 39.1, 27.0 and 33.9 wt% of bassanite, aragonite and gibbsite, respectively, but much closer to the RQPA results reported in Table 4. We speculate that this could be due to several reasons including that the amorphous $Al(OH)_3$ gel may contain calcium species and not full decomposition of ettringite according to reaction (3). More research is needed to clarify this disagreement. Finally, it is worth noting that the low r-value PDF traces, between 1.5 and 10 Å, for these two materials are well justified with the crystalline and nanocrystalline components. Therefore, the amount of any amorphous phase must be quite small. It is highlighted here that the typical broad bands at ~4.0 and 6.1 Å of ACC, see below, is not evident in the difference curve of Figure 6d.

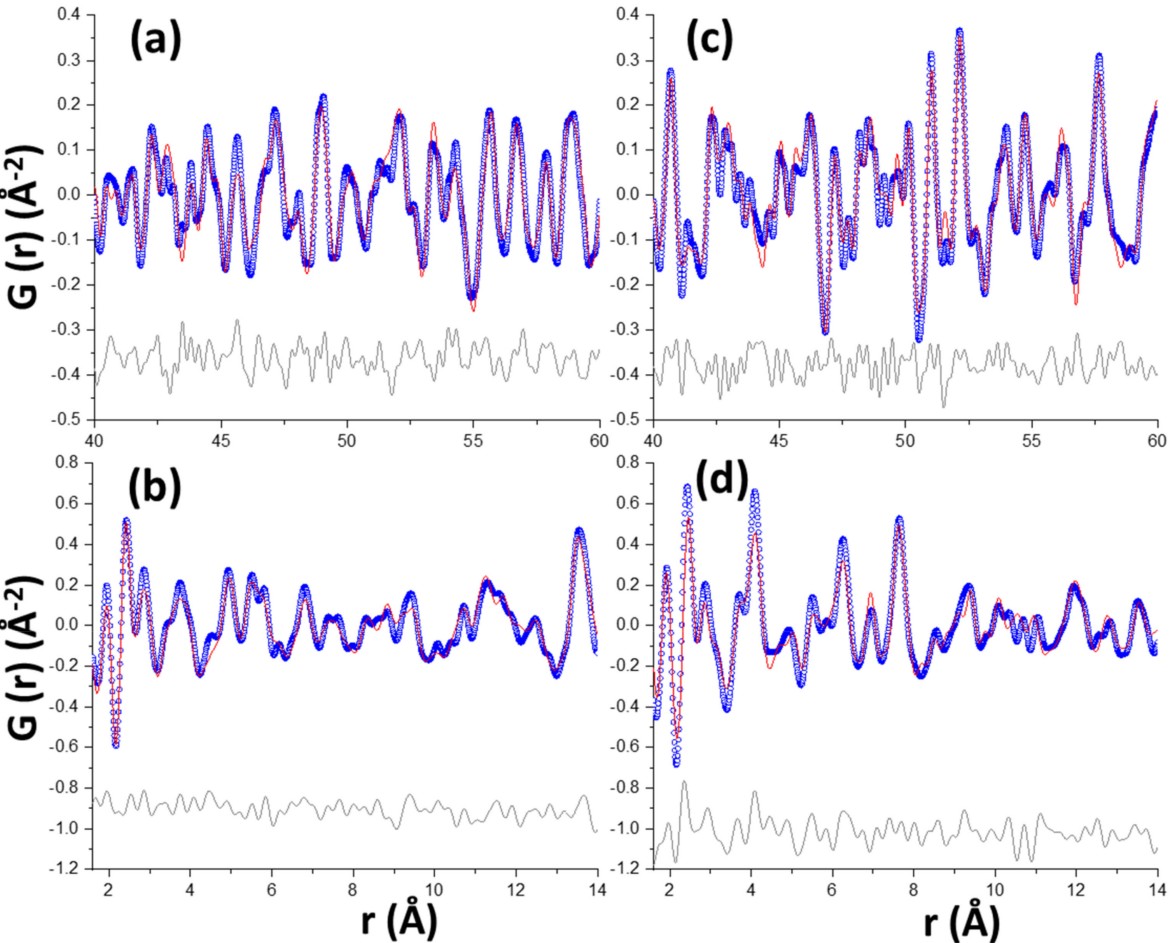

**Figure 6.** Experimental (blue circles) and fitted (red solid line) PDF patterns for Y-A_hyd: (**a**,**b**) panels and Y-A_carb: (**c**,**d**) panels. Difference curves are shown as grey lines at the bottom of each panel.

The PDF analysis of the $C_3S$ and $C_2S$ pastes are more complex as the amount of nanocrystalline and amorphous phases are larger and their structural descriptions are still not well established. Figure 7 displays the high r-range results, i.e., the crystalline phases contributions, for these samples. The fit for $C_3S\_hyd$ is shown in Figure 7a and the pattern is well justified just with portlandite, $R_W$ = 23.3%. The similar analysis for $C_3S\_carb$, $R_W$ = 19.1%, gave 44, 30 and 26 wt% for aragonite, vaterite and calcite, respectively; see Figure 7b. The fit for $C_2S\_hyd$ is shown in Figure 7c and it was the poorest with $R_W$ = 42.4%, indicating problems in the fit are likely due to additional scattering contributions not accounted for. Every attempt to improve this refinement was unsuccessful. The final fit yielded 57, 38 and 5 wt% of β-$C_2S$, γ-$C_2S$ and CH, respectively. The corresponding analysis for $C_2S\_carb$ was very good, $R_W$ = 18.5%, resulting in 48, 36 and 16 wt% for aragonite, calcite and β-$C_2S$, respectively; see Figure 7d.

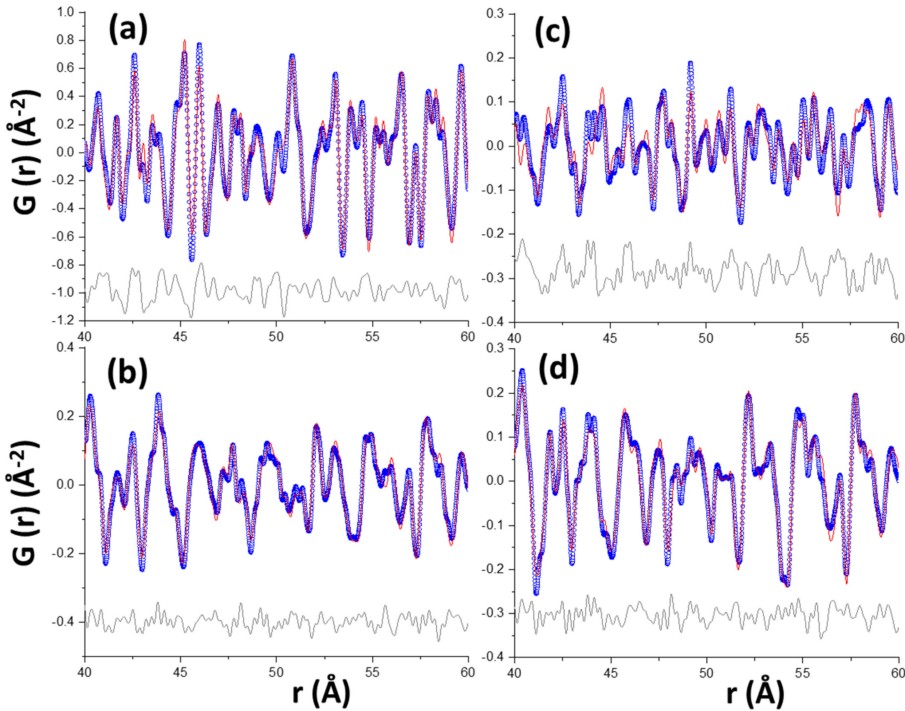

**Figure 7.** Experimental (blue circles) and fitted (red solid line) large r-range PDF patterns for (**a**) $C_3S\_hyd$, (**b**) $C_3S\_carb$, (**c**) $C_2S\_hyd$ and (**d**) $C_2S\_carb$. Difference curves are shown as grey lines at the bottom of each panel.

Now, we turn our attention to the low r-ranges. Due to the lack of structural descriptions for nanocrystalline and amorphous components, we have employed the differential PDF approach [23,32,53]. This methodology consists on subtracting from the raw patterns, some known contributions, in this case the ones from the crystalline components. For doing this, the parameters for the crystalline phases (scale, unit cell and atomic displacement parameters) that were refined in the 40–60 Å range, are kept fixed in the low PDF region and the difference/subtracted curve corresponds to the contribution of non-crystalline phases (amorphous and nanocrystalline). Hence, Figure 8 shows the low r-range differential PDF patterns for $C_3S\_hyd$, $C_3S\_carb$, $C_2S\_hyd$ and $C_2S\_carb$. These PDF traces for $C_3S\_hyd$ and $C_2S\_hyd$, Figure 8a,c, are consistent with the contribution of C-S-H gel and a few layer thick nanoportlandite. This has been extensively discussed very recently and the interested reader is addressed to the publication [23]. The changes in the differential PDF traces for the carbonate materials are readily evident. Very importantly, the key interatomic correlation at ~3.7 Å (which corresponds to Ca⋯Ca and Ca⋯Si) in $C_3S\_hyd$ and $C_2S\_hyd$, highlighted with broken lines in Figure 8, is almost vanished in $C_3S\_carb$ and $C_2S\_carb$.

This indicated that almost all C-S-H gel in these samples has been carbonated to yield (different forms of) $CaCO_3$ and (probably calcium-containing) $SiO_2 \cdot nH_2O$.

The crystalline products of carbonation reactions (1) and (2) have been already accounted for as detailed above. The products of reactions (1) and (2) that can be amorphous are silica and calcium carbonate. On the one hand, the synchrotron PDF pattern of amorphous silica has been reported [13,54,55]. One of the most intense interatomic correlations is located at 1.62 Å and it is due to the Si-O bonds, see Figure 8b,d. There are three additional, less intense peaks, at 2.6, 3.0 and 4.1 Å due to O⋯O, Si⋯Si and second Si⋯O interatomic correlations, respectively. The interatomic distance peak of O⋯O is not labelled in Figure 8 due to its low intensity. The interatomic distance at 1.62 Å is evident but it is typical of any silicate and therefore it cannot be used to disentangle the amorphous silica contribution. However, the second most intense correlation, the Si⋯Si distance at 3.0 Å can be used to follow amorphous silica development. This band is partly overlapped with Ca⋯C and Ca⋯O correlations as labelled in Figure 8b,d. Interestingly, $C_2S\_carb$ has a large Si/Ca ratio (due to its stoichiometry) and the band due to the Si⋯Si interatomic correlation is more intense. Additionally, there is scattering in the remaining two real-space distances, at 2.6 and 4.1 Å, but the low intensity of the signals does not allow to firmly establish their origin. In any case, although the differential PDF traces do not allow us to unequivocally demonstrate the existence of amorphous silica, the atomic correlations are fully in line with its presence as previously reported for the carbonation of a Portland cement paste [13,56].

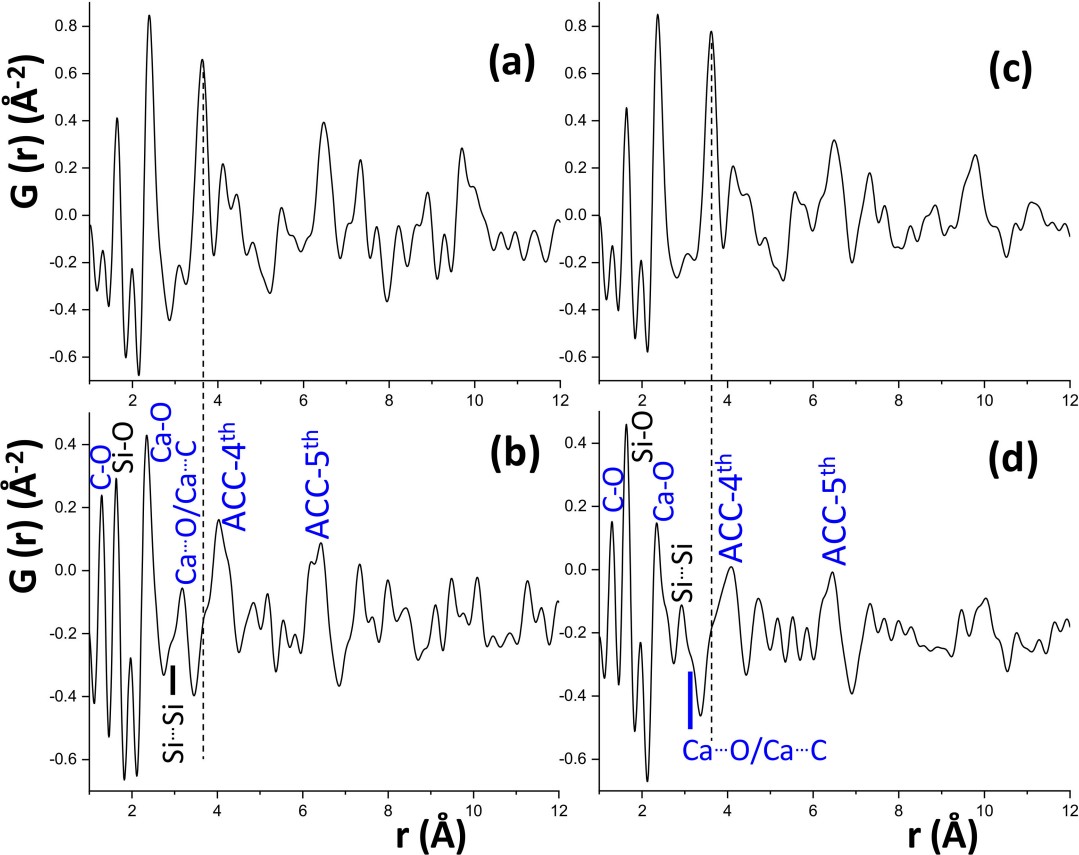

**Figure 8.** Low r-range differential PDF patterns for (**a**) $C_3S\_hyd$, (**b**) $C_3S\_carb$, (**c**) $C_2S\_hyd$ and (**d**) $C_2S\_carb$. These traces display the difference between the raw PDFs patterns and the calculated contributions from the crystalline contents through the fits to the high r-range region as displayed in Figure 7. Therefore, these are the contributions of the nanocrystalline and amorphous phase contents. Prominent peaks due to amorphous silica (black) and amorphous calcium carbonate (blue) are labelled for the carbonated samples.

On the other hand, the PDF traces of ACC have been extensively studied due to their importance in the biogenesis of calcium carbonate biominerals [38,39]. Although the contributions of the crystalline calcium carbonates have already been subtracted, Figure 8b,d shows intense (sharp) bands at 1.3 and 2.4 Å typical of any $CaCO_3$: C-O and Ca-O bonds, respectively. This clearly points toward the presence of an ACC in the carbonated samples. Moreover, in addition to the low intensity band at 3.2 Å, the PDF traces of $C_3S\_carb$ and $C_2S\_carb$ display two broad bands at ~4.0 and 6.1 Å. All these peaks, but especially the two broad ones, are the signature of the presence of ACC. Therefore, the presence of amorphous calcium carbonate in $C_3S\_carb$ and $C_2S\_carb$ is firmly established from the recorded PDF data by using the differential approach. However, as there is not available structural description, its quantification is still not possible.

### 3.3. Thermal Analysis

Figure 9 shows the thermal analysis results for Y-A_hyd and Y-A_carb. For Y-A_hyd sample, the nominal water added (w/c = 0.85) should result in an overall weight loss of 46.0%. However, 51.0 is observed and we justify this larger weight loss because the nanocrystalline gibbsite gel can (physically) adsorb water. As expected, carbonation does not take place in this sample, and the absence of a significant weight loss in the 550–800 °C interval is seen. For the nominal (expected) amount of nano-gibbsite according to reaction (5), 19.9%, should result in an expected weight loss of 7.2%. The measured weight loss in the 175–300 °C temperature range was 8.3%, slightly larger than expected and in line with the larger nano-gibbsite content measured by PDF and RQPA analyses, see above.

For Y-A_carb sample, the measured contents from PDF were 34.1, 20.7 and 45.2% for bassanite, aragonite and nano-gibbsite, respectively. The amounts expected from the stoichiometric hydration, followed by the carbonation reaction (3) are: 39.1, 27.0 and 33.9, respectively. These three phases have their respective weight losses that must be normalised taking into account their contents. Hence, the theoretically expected weight losses are 2.4, 11.8 and 11.7% for bassanite, aragonite and nano-gibbsite, respectively. Again, the overall measured mass loss, 34.3% is larger than the sum, 25.8%, likely due to the physically absorbed water in the nanocrystalline gel. The carbonate measured weight loss, 9.2%, is smaller than the expected one, 11.8%, pointing towards partial presence of calcium within the $Al(OH)_3$ gel as previously reported [57]. Moreover, it is not possible to rule out the presence of a small proportion of ACC and/or amorphous $Al(OH)_3$ gel in the carbonated sample. The thermal analysis trace in Figure 9 (bottom) shows a 2.1% of weight loss in the 550–650 °C range, which could be related to the decarbonation of ACC. The mass loss in the 160–300 °C temperature range was 10.4%, slightly less than the expected value, 11.7%. A larger difference is found in the lower temperature range as it could not disentangle the weight loss from bassanite due to the physically absorbed water in the gel. Finally, it is worth noting that the calcium carbonate decomposition is a single endotherm centred at 730 °C.

Concerning the thermal decomposition features of PC pastes, Thiery et al. [58] made two key contributions. On the one hand, using mass spectrometry, they demonstrated that the mass losses above 500 °C were due to $CO_2$ release and not water. On the other hand, for the calcium carbonate decomposition, they found and reported three modes from the differential TG traces, all coexisting in the ultimate state of carbonation of PC binders. mode I (780 °C < T < 990 °C in their samples) was supposedly associated with the decomposition of well-crystallised calcite. The presence of vaterite and aragonite may cause the carbonates to decompose at lower temperatures, and it could define mode II (700 °C < T < 780 °C). Finally, mode III (550 °C < T < 700 °C) was tentatively to be associated with amorphous calcium carbonate.

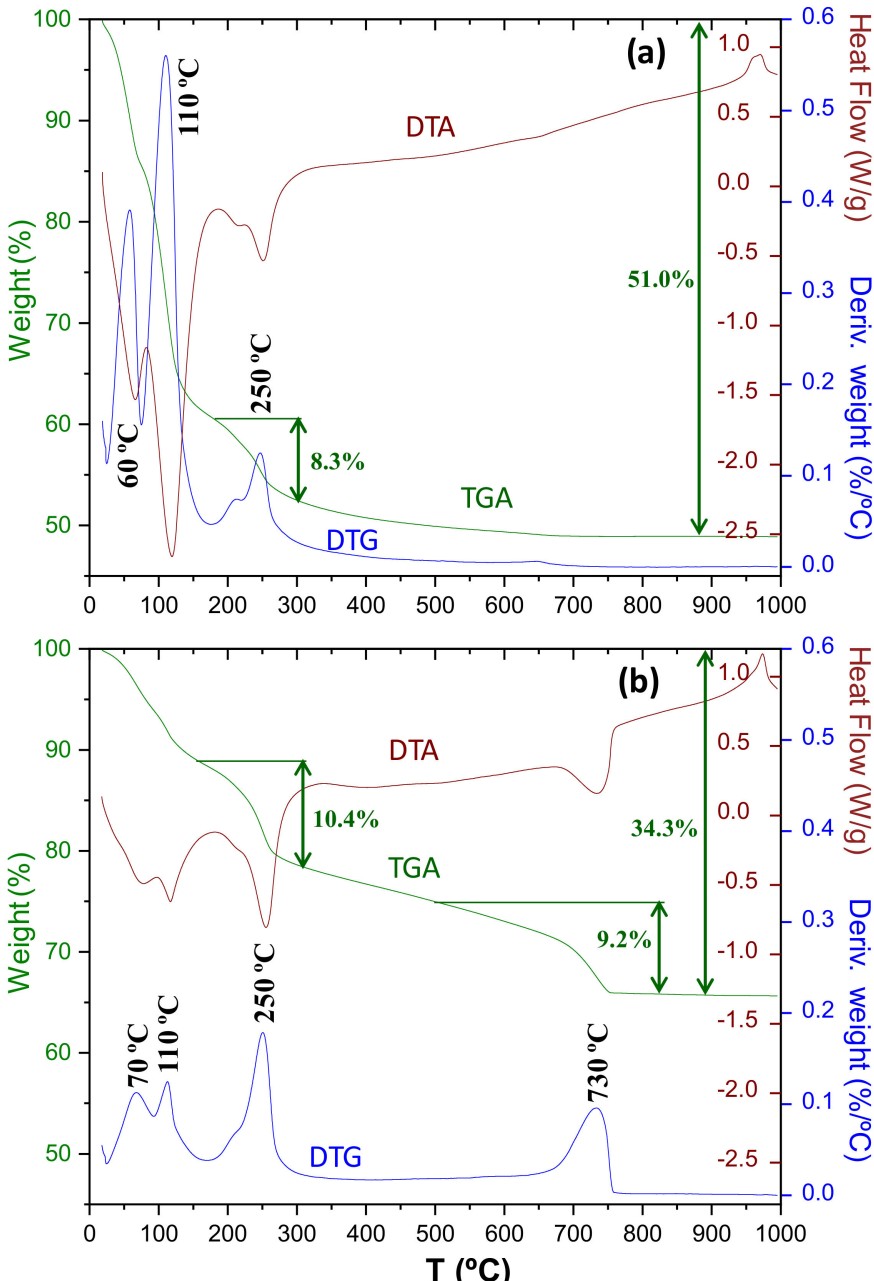

**Figure 9.** Thermal analysis traces (green: weight losses, blue: derivative of the weight losses and brown: heat flow) for (**a**) Y-A_hyd, and (**b**) Y-A_carb. For Y-A_hyd, the reported weight losses 8.3 and 51.0% were calculated from the temperature ranges 175–300 and RT–900 °C, respectively. For Y-A_carb, the reported weight losses 10.4, 9.2 and 34.3% were calculated from the temperature ranges 160–310, 500–825 and RT–900 °C, respectively.

Figure 10 shows the thermal analysis results for $C_3S$_hyd and $C_3S$_carb. For $C_3S$_hyd sample, the nominal water added (w/c = 0.50) should result in an overall weight loss of 33.3% which agrees very well with the measured value, 32.9%. Furthermore, carbonation has not taken place, even as ACC, as there is no significant weight loss above 550 °C. The weight loss due to the portlandite, close to 450 °C, was determined from the slope method [59] not taking into account the loss due to C-S-H in the same temperature range. The measured value 5.6% is slightly smaller than the expected value according to chemical reaction (6), 6.7%. However, we consider the agreement satisfactory given the approxima-

tions in the stoichiometry of the reaction and the way in which the experimental value is determined, i.e., the weight loss of this sample between 400 and 550 °C is 7.0%.

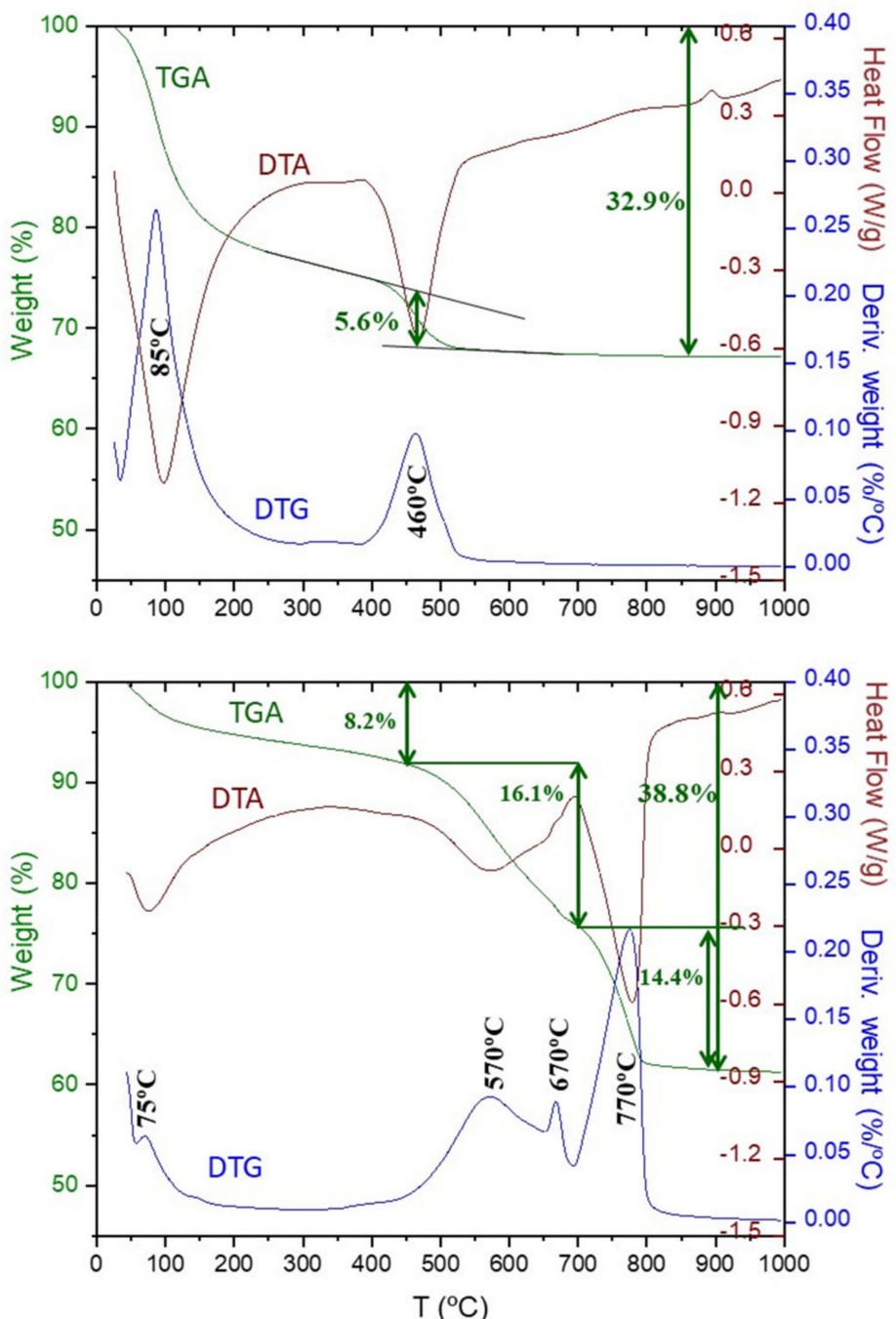

**Figure 10.** Thermal analysis traces as in Figure 9 for (**a**) $C_3S\_hyd$ and (**b**) $C_3S\_carb$. For $C_3S\_hyd$, the reported weight losses 5.6 and 32.9% were calculated from the slope method centred at 460 °C (to correct from the C-S-H contribution at these temperatures) and RT–900 °C, respectively. For $C_3S\_carb$, the reported weight losses 8.2, 16.1, 14.4 and 38.8% were calculated from the temperature ranges RT–450, 450–700, 700–900 and RT–950 °C, respectively.

For $C_3S\_carb$ sample, the first weight loss between RT and ~450 °C, 8.2% is assigned to the water content of the silica gel and the (likely small) fraction of remaining C-S-H gel. Moreover, the presence of a small amount of free water cannot be discarded in this region. In agreement with the Rietveld content where ~0.6 wt% of Portlandite

was measured, no weight loss associated to this phase is observed in the thermal traces. Above 500 °C two endotherms are observed close to 570 and 770 °C. In the derivative of the weight loss, three peaks of different widths are measured at 570, 670 and 770 °C. These three peaks (three $CaCO_3$ decomposition modes) have been widely reported in the cement carbonation bibliography [12,14,58,60,61]. The broad endotherm centred at ~570 °C, mode-III, is generally associated to the mass loss from calcium carbonate from the C-S-H gel [12,14,58], which is measured here as 16.1%. This weight loss also contains the mode-II contribution as they are severely overlapped. The relatively sharper endotherm centred here at ~770 °C, mode-I, is associated to the mass loss from the crystalline calcium carbonates, measured as 14.4%. It is noted here that the current accepted knowledge is that both vaterite and aragonite transform on heating to calcite at variable temperatures close to 400–500 °C [62,63], and therefore, the measured loss in the 700–900 °C range could correspond to the whole amount of crystalline carbonates. Furthermore, it is currently not known if the ACC phases, in the carbonated pastes, crystallise on heating. A back of the envelop calculation concerning $CaCO_3$ for $C_3S\_carb$ would suggest 1.8 moles from C-S-H and 1.2 moles from CH according to chemical reaction (6). Hence, assuming full carbonation of both phases (C-S-H to give carbonates with weight losses in the 450–700 °C temperature range, modes II and III, and CH to give crystalline carbonates with weight losses at 700–900 °C, mode I) the mass losses should be 60% and 40%, respectively. The experimental values are 53% and 47% showing a relatively good agreement for such unsophisticated calculation. On the other hand, the weight loss for mode-III is too large for being assigned to just amorphous calcium carbonate decomposition. Indeed, more research is needed but the results reported here challenge a commonly accepted understanding that all crystalline $CaCO_3$ decompose above 700 °C.

Figure 11 shows the thermal analysis data for $C_2S\_hyd$ and $C_2S\_carb$. For $C_2S\_hyd$ sample, the nominal water added (w/c = 0.45) should result in an overall weight loss of 31.0% which agrees fairly well with the measured value, 32.8. This sample is also not carbonated from the atmosphere as the weight loss above 550 °C is negligible. The weight loss due to the portlandite, close to 420 °C, determined from the slope method was 0.6%. This value is smaller than the expected value according to reaction (7), 1.5%. However, this difference could arise from: (i) The belite hydration degree being lower that 100% as measured by diffraction; (ii) the experimental errors; and (iii) because the calcium content of the C-S-H gel can be slightly higher than 1.80 as previously reported [23].

For $C_2S\_carb$ sample, the first weight loss between RT and ~450 °C, 11.2% is assigned to the water content of the silica gel and also the possible (small) fraction of remaining C-S-H gel. It is underlined here the internal consistency of the obtained results as $C_2S\_carb$ contains more silica gel content than $C_3S\_carb$ and the weight losses in this range were 11.2 and 8.2%, respectively. As expected, no losses from portlandite is observed in this sample. Above 500 °C, again two endotherms are observed close to 600 and 740 °C, corresponding to modes-III and I, respectively. It is underlined that again mode-II is clearly seen in the DTG trace but not in the DTA one, which suggests that the thermal exchange of that process is very small. From DTG curve and for this sample, $CaCO_3$ decomposition modes-III –II and –I have different widths and are located at 600, 675 and 740 °C, respectively. Mode-III is a very broad endotherm centred at ~600 °C which has an associated mass loss of 18.5%. This mass loss again contains the mode-II contribution as both effects are strongly overlapped. Mode-I is a broad endotherm centred at ~740 °C which has an associated mass loss of 6.0%. The ratio between modes III-II and mode I is 75/25. This is in semiquantitative agreement with our hypothesis that the first ($CaCO_3$-due) weight loss comes from the C-S-H gel carbonation but not in quantitative agreement as even a lower ratio would be expected for this sample close to 90/10. More research is needed but although the ACC presence is firmly established by differential PDF, the ratio amorphous-crystalline $CaCO_3$ cannot be 3-to-1 as derived from the TA study if the weight loss from 450–700 °C is considered as arising from the amorphous calcium carbonate content.



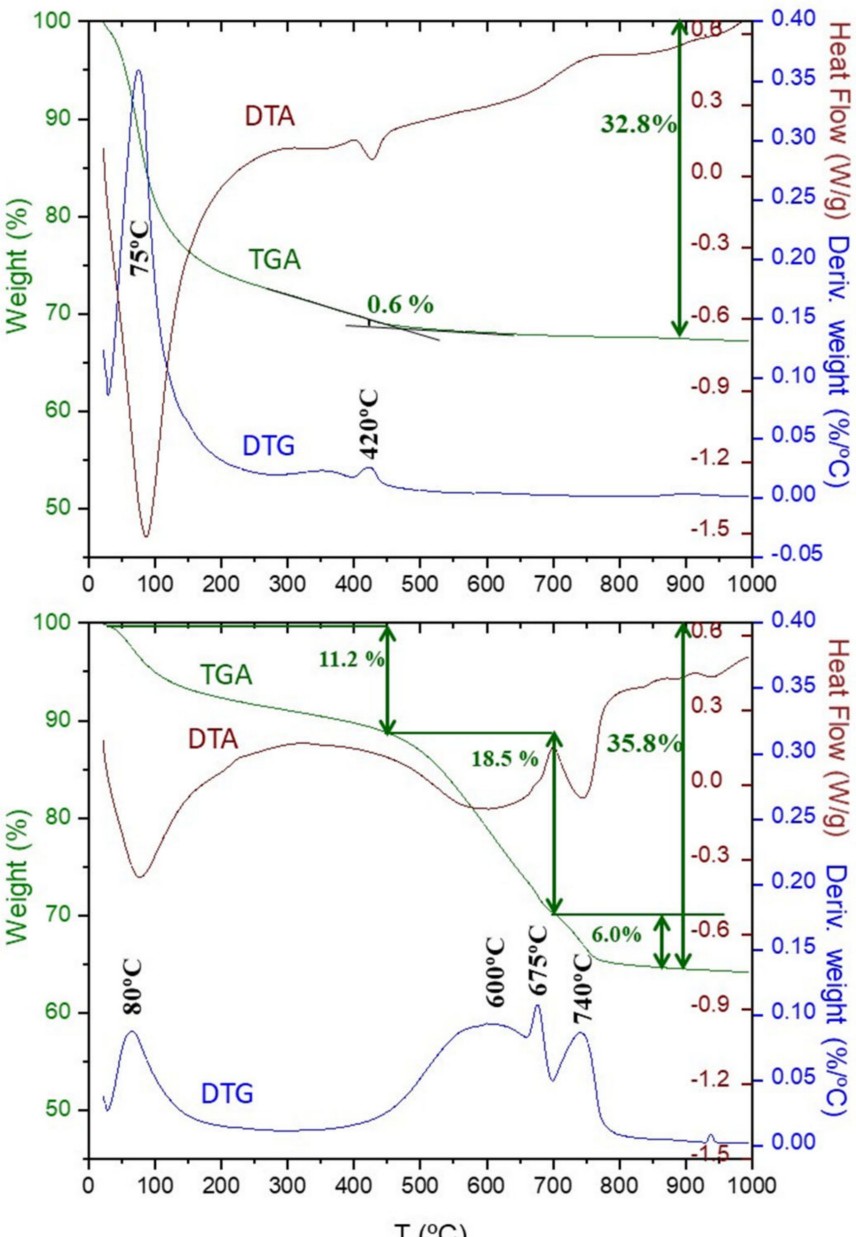

**Figure 11.** Thermal analysis traces as in Figure 9 for (**a**) C$_2$S_hyd and (**b**) C$_2$S_carb. For C$_2$S_hyd, the reported weight losses 0.6 and 32.8% were calculated from the slope method centred at 420 °C (to correct from the C-S-H contribution at these temperatures) and RT–900 °C, respectively. For C$_2$S_carb, the reported weight losses 11.2, 18.5, 6.0 and 35.8% were calculated from the temperature ranges RT–450, 450–700, 700–900 and RT–950 °C, respectively.

## 4. Conclusions

From the data reported in this work, the following main conclusions can be drawn for the studied reference pastes.

- The reaction of ye'elimite with the stoichiometric amount of anhydrite yields crystalline ettringite and gibbsite with nanocrystalline nature (average particle size ~3 nm). After accelerated carbonation (3%CO$_2$ and RH = 65%), bassanite, aragonite and gibbsite are formed and quantified. The particle size of gibbsite under these conditions was larger, ~5 nm. Chiefly, PDF analysis indicated that any amorphous content should be low and no appreciable amounts of amorphous calcium carbonate are formed in these experimental conditions.

- Crystalline CaCO3 polymorph contents (vaterite, aragonite and calcite) were highly variable in the three studied samples and we still cannot rationalise the conditions for their formation.
- C-S-H carbonates give: (i) amorphous silica gel; (ii) amorphous calcium carbonate, and (iii) variable contents of crystalline calcium carbonates. The existence of both coexisting amorphous components has been firmly established from the differential PDF study, being the most relevant and novel result from this work.
- Carbonate decomposition in the 500–700 °C temperature range (modes II and III) is intimately joined to C-S-H carbonation. Larger C-S-H content led to larger contribution of the II and III modes. Modes-II and III contain the weight loss from amorphous calcium carbonate but they should also have a contribution from metastable crystalline calcium carbonates (vaterite and aragonite).

**Author Contributions:** A.C.: conceptualization, methodology, formal analysis, investigation, data curation, writing—original draft, supervision, funding acquisition. A.G.D.l.T.: formal analysis, investigation, writing—review and editing. M.A.G.A.: conceptualization, investigation, writing—original draft. All authors have read and agreed to the published version of the manuscript.

**Funding:** This research was funded by Ministry of Science (Spain), grant number PID2019-104378RJ-I00, and Junta de Andalucía (Spain), grant number P18-RT-720.

**Data Availability Statement:** All raw data used in this article (synchrotron patterns and thermal traces) can be freely accessed on Zenodo at https://doi.org/10.5281/zenodo.4607315 (accessed on 13 May 2021), and used under the Creative Commons Attribution license.

**Acknowledgments:** We thank CELLS-ALBA (Barcelona, Spain) for providing synchrotron beam time at BL04-MSPD and Oriol Vallcorba for his assistance during the experiment. We thank Maurizio Marchi from Italcementi (HeidelbergCement Group) for performing the accelerated carbonation on the hydrated samples.

**Conflicts of Interest:** The authors declare no conflict of interest.

## Appendix A

**Table A1.** Crystal structures used in the Rietveld quantitative phase analysis including the ICSD numbers.

| Phase Name | ICSD | Ref. |
|---|---|---|
| t-$C_3S$ | 4331 | [64] |
| β-$C_2S$ | 81096 | [65] |
| γ-$C_2S$ | 81095 | [65] |
| CH | 202220 | [66] |
| AFt | 155395 | [67] |
| Calcite | 80869 | [68] |
| Vaterite | 15879 | [69] |
| Aragonite | 157994 | [70] |
| Bassanite | 79529 | [71] |
| Monocarbonate | 59327 | [72] |
| nano-Gibbsite | 6162 | [73] |

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
