# Peer review of "X-ray Total Scattering Study of Phases Formed from Cement Phases Carbonation"

_minerals, doi:10.3390/min11050519_

Round 1
Reviewer 1 Report
The reviewed manuscript reports the study of phase changes occurring during the cement phase carbonation. Interesting and valuable research results were presented that may allow better understanding of cement composite carbonation effects. They were supported with a thorough literature review.
However, this paper requires improvement and some clarification. The quality of translation into English raises questions. The syntax is not always correct and can affect correct understanding. There are also other errors - e.g. punctuation. Selected examples of the linguistic errors will be indicated below. In general, I recommend that you review the entire text to ensure that it reflects the authors' intentions well.
Some of the tests, in particular thermal analysis, also require additional clarifications.
Below is a detailed list of comments:
1. Title. It is incorrected and imprecise. It suggests that the study focuses only on the amorphous phase, which is not true - different forms of CaCO3 were tested. I suggest you rethink its wording and replace it with, for example: X-ray total scattering study of phases formed as a result of cement phases carbonation.
2. l.34-35: “There is plenty of research in carbonation of building materials and they cannot be discussed here” - this sentence is incorrectly composed. I don't entirely agree with this wording. This paper deals with the study of the carbonation of building materials. I understand that the Authors meant that their research did not focus on the impact of carbonation on building materials performance properties, but only on the phase changes. For this reason, in l.96-110, they cite only two works [13, 33] referring strictly to the use of PDF for the interpretation of carbonation process research. I suggest that the Authors add examples of papers where the results of their study could be applied to interpret the results – e.g. Bae S. et Al. Construction and Building Materials 237, 2020, 117714 or Kupwade-Patil K. Cement and Concrete Research 126, 2019, 105920. This would highlight the significance of this work.
3. Table 1 and 3. References to the literature for the formulas used must be provided. For example, the C-S-H phase formula is debatable - contrary to what the authors write, there is no universal agreement for the coefficient x~1.7-1.8 (l.81), therefore the source must be given. In the light of available literature, this value can be considered relatively low (Kurdowski W., Cement and Concrete Chemistry, Springer 2014 or Richardson I., Taylor H.F.W., Cement Chemistry, ICE Publishing 2017). This is evidenced by the authors' own observations contained in the study - l.459. It must be remembered that the C-S-H phase is not formed in a stoichiometric reaction and that different types of C-S-H phase exist.Moreover, the use of the same "n" and "y" coefficients in the carbonation formula for the C-S-H and AFt phases raises doubts.
In l.262 the Authors mention the AFm phase (“AFm – type phase of monocarbonate”). I believe that in such case, it should also be mentioned in the introduction, because, like AFt, AFm phase can undergo the carbonation process. A reference to the literature should also be provided here.
4. Chemical formulas are misspelled, eg. "Ca(OH)2" - should be "Ca(OH)2" – remember to use subscripts in the formula.
5. Section 2. I believe that the layout of the subsections must be changed. The PSD and BET studies were described (section 2.7 and 2.8) after their results were reported (section 2.1). I advise to move subsections 2.1 and 2.2 to the end of this section. Rietvield and PDF analyzes are methods of the X-ray diffraction tests interpretation - they should be included in subsections 2.3.1 and 2.3.2, respectively.
6. l.30:„”- wrong symbol.
7. l.137-154. Please better specify the tested samples exact age, because readers may have some doubts about the age of non-carbonated and carbonated samples. What was the storage time for samples in the vacuum sealed bag? Was the storage time of non-carbonated samples extended by 80 days, while the remaining samples were carbonated (this is suggested by Fig. 1)? Should it be understood that "reference carbonated samples"(l.147) were not carbonated? - This is from the description and is misleading.
8. l.174: “1 to 130° (2θ)” - this is a problematic formulation. The authors do not present full results for this scale. In Figure 2, "Å" was used in place of "°", and other figures used a smaller scale. The descriptions are not consistent.
Is there a need to present the diffraction patterns for 2θ > 60° at all? – they only reduce the readability of fig. 2.
Did the measurement really start from 1°? - it is not visible on the X-ray patterns.
9. Section 2.6-2.8. More detailed information on the performed tests should be provided. This is especially important for thermal analysis as entire section 3.3 is devoted to thermal analysis results.
- how were the samples for these tests prepared? - very large mass losses at temperatures below 100 °C, especially for non-carbonated samples, suggest that they may have been damp. In such a case, the presence of free water should be taken into account when making the calculations. In section 2.1. the Authors do not mention that the samples were dried, so I do not consider it correct to combine the total weight loss to 200 °C only with the evaporation of water from “silica gel and the C-S-H phase” (l.428-429 and 460-461).
What do the Authors understand by the term "synthetic air" (l.295)? What was the composition of this air? What was the gas flow speed? Why was nitrogen not used?
10. All abbreviations should be explained, eg "RT" (first appearance in l.194), "RQPA" (l.220 - referring to the subsection title it should be SRQPA?).
11. Fig 3-5. For easier comparison, the same scale should be used for the Y-axis in all X-ray patterns.
12. l.338-354 - the description in the text is inconsistent with what can be seen in Figure 8, e.g.:
- l. 341-341: “The most intense interatomic correlation is located at 1.62 Å and it is due to the Si-O bonds.” - in figure 8 for 1.62 Å, the C-O bonds were marked, not Si-O. In Fig. 8b, Ca-O is more intense.
- l. 343: “…less intense, peaks, at 2.6, 3.0 and 4.1 Å due to O···O…” - in Figure 8, the O···O peak was not marked at all.
13. l 263: “When this sample is carbonated, several results can be drawn” – sentence incorrect.
14. l.387: „bassanite aragonite” - lack of comma.
15. l.29: „[4,5]” – font is too large.
Author Response
Please see the attachment for full letter
---------------------------
Comments from the reviewers:
Reviewer #1
The reviewed manuscript reports the study of phase changes occurring during the cement phase carbonation. Interesting and valuable research results were presented that may allow better understanding of cement composite carbonation effects. They were supported with a thorough literature review.
However, this paper requires improvement and some clarification. The quality of translation into English raises questions. The syntax is not always correct and can affect correct understanding. There are also other errors - e.g. punctuation. Selected examples of the linguistic errors will be indicated below. In general, I recommend that you review the entire text to ensure that it reflects the authors' intentions well. Some of the tests, in particular thermal analysis, also require additional clarifications.
Author reply: We thank him/her for the time and effort invested to clarify (and therefore improve) the manuscript. We have edit the English to our best and several points are addressed in the sections below.
Below is a detailed list of comments:
R1.1. Title. It is incorrected and imprecise. It suggests that the study focuses only on the amorphous phase, which is not true - different forms of CaCO3 were tested. I suggest you rethink its wording and replace it with, for example: X-ray total scattering study of phases formed as a result of cement phases carbonation.
Author reply: We have followed the referee’s indication. We have modified the title according his/her remark. Now the title is: “X-ray total scattering study of phases formed from cement phases carbonation”
R1.2. l.34-35: “There is plenty of research in carbonation of building materials and they cannot be discussed here” - this sentence is incorrectly composed. I don't entirely agree with this wording. This paper deals with the study of the carbonation of building materials. I understand that the Authors meant that their research did not focus on the impact of carbonation on building materials performance properties, but only on the phase changes. For this reason, in l.96-110, they cite only two works [13, 33] referring strictly to the use of PDF for the interpretation of carbonation process research. I suggest that the Authors add examples of papers where the results of their study could be applied to interpret the results – e.g. Bae S. et Al. Construction and Building Materials 237, 2020, 117714 or Kupwade-Patil K. Cement and Concrete Research 126, 2019, 105920. This would highlight the significance of this work.
Author reply: This point has also been addressed by the second reviewer (remark 2.4) and therefore, it is carefully taken care of here.
On the one hand, and as correctly noted by the reviewer, we have replaced “There is plenty of research in carbonation of building materials and they cannot be discussed here.” By “There is plenty of research in carbonation of building materials and this investigation focusses on the phase changes and not on the impact of carbonation on building materials performance properties."
On the other hand, and also answering the reviewer recommendation, we have enlarged the discussion in the introduction by adding some works using PDF approach to address/solve relevant issues in the cement chemistry field. Now in the manuscript: “PDF methodology has been recently used for improving our understanding in cement chemistry. In situ synchrotron and neutron PDF were employed to study the hydration of alite and G-type oil well cement including the role of a phosphonate retarder [34]. It was shown that calcium complexation by the retarder was key, with C-S-H gel nuclei poisoning and CH partial precipitation inhibition also contributing. In situ synchrotron PDF was also employed to study the atomic structural deformation of C-S-H under external loading. The consequences of applied loading were measured by: i) the strain gauge, ii) the d-spacing shifts (reciprocal space), and the interatomic distance shifts (real space) [35]. For the r < 20 Å range (in real space), where the C-S-H contribution dominates, a 53-year-old alite paste had a much higher overall elastic modulus than a 131-day-old paste, being 18.3 and 8.3 GPa, respectively. PDF can also be used to investigate alternative sustainable binders which may contain more alkalis [36]. Many sodium-substituted calcium-(alumino-)silicate-hydrate gels were studied and it was shown that the addition of higher levels of alkalis resulted in a systematic reduction of the degree of silicate polymerization of the gels with clear consequences in their nanoscale ordering.” (lines 99-114)
[34] Kupwade-Patil, K.; Boul, P.J.; Rasner, D.K.; Everett, S.M.; Proffen, T.; Page, K.; Ma, D.; Olds, D.; Thaemlitz, C.J.; Büyüköztürk, O. Retarder effect on hydrating oil well cements investigated using in situ neutron/X-ray pair distribution function analysis. Cem. Concr. Res. 2019, 126, 105920.
[35] Bae, S.; Jee, H.; Suh, H.; Kanematsu, M.; Shiro, A.; Machida, A.; Watanuki, T.; Shobu, T.; Morooka, S.; Geng, G.; et al. Analysis of atomistic structural deformation characteristics of calcium silicate hydrate in 53-year-old tricalcium silicate paste using atomic pair distribution function. Constr. Build. Mater. 2020, 237, 117714.
[36] Garg, N.; Özçelik, V.O.; Skibsted, J.; White, C.E. Nanoscale Ordering and Depolymerization of Calcium Silicate Hydrates in the Presence of Alkalis. J. Phys. Chem. C 2019, 123, 24873–24883.
R1.3. Table 1 and 3. References to the literature for the formulas used must be provided. For example, the C-S-H phase formula is debatable - contrary to what the authors write, there is no universal agreement for the coefficient x~1.7-1.8 (l.81), therefore the source must be given. In the light of available literature, this value can be considered relatively low (Kurdowski W., Cement and Concrete Chemistry, Springer 2014 or Richardson I., Taylor H.F.W., Cement Chemistry, ICE Publishing 2017). This is evidenced by the authors' own observations contained in the study - l.459. It must be remembered that the C-S-H phase is not formed in a stoichiometric reaction and that different types of C-S-H phase exist.
Author reply-I: We have included the following references for Table 1 and 3. “Table 1. Carbonation reactions of selected components in hydrated cement binders [7].” and “Table 3. Stoichiometries of the hydration reactions for the phases employed in this study [31,32,46].”
[7] Šavija, B.; Lukovic, M. Carbonation of cement paste: Understanding, challenges, and opportunities. Constr. Build. Mater. 2016, 117, 285–301.
[31] Cuesta, A.; Zea-Garcia, J.D.; Londono-Zuluaga, D.; De la Torre, A.G.; Santacruz, I.; Vallcorba, O.; Dapiaggi, M.; Sanfélix, S.G.; Aranda, M.A.G. Multiscale understanding of tricalcium silicate hydration reactions. Sci. Rep. 2018, 8, 8544.
[32] Cuesta, A.; Santacruz, I.; De la Torre, A.G.; Dapiaggi, M.; Zea-Garcia, J.D.; Aranda, M.A.G. Local structure and Ca/Si ratio in C-S-H gels from hydration of blends of tricalcium silicate and silica fume. Cem. Concr. Res. 2021, 143, 106405.
[46] Cuesta, A.; Álvarez-Pinazo, G.; Sanfélix, S.G.; Peral, I.; Aranda, M.A.G.; De la Torre, A.G. Hydration mechanisms of two polymorphs of synthetic ye’elimite. Cem. Concr. Res. 2014, 63, 127–136.
Moreover, we agree that the Ca/Si coefficient could be higher than 1.8, we have also pointed out this in comment R2.28 by adding a reference. We have also modified the footnote in table 1 and adding some references: “* Calcium silicate hydrate gel composition is variable but close to x~1.7-1.9 and n~4.0 for neat Portland cement pastes [23,24].”
[23] Cuesta, A.; Santacruz, I.; De la Torre, A.G.; Dapiaggi, M.; Zea-Garcia, J.D.; Aranda, M.A.G. Local structure and Ca/Si ratio in C-S-H gels from hydration of blends of tricalcium silicate and silica fume. Cem. Concr. Res. 2021, 143, 106405.
[24] Taylor, H.F.W. Cement chemistry; 2nd ed.; Thomas Telford Pub: London, UK, 1997; ISBN 9780727725929.
Moreover, the use of the same "n" and "y" coefficients in the carbonation formula for the C-S-H and AFt phases raises doubts.
Author reply-II: The referee is right. We have changed the coefficients for AFt, now: “AFt + 3CO2 ® 3CaCO3 + 3CaSO4·mH2O + 2Al(OH)3 + zH2O”
In l.262 the Authors mention the AFm phase (“AFm – type phase of monocarbonate”). I believe that in such case, it should also be mentioned in the introduction, because, like AFt, AFm phase can undergo the carbonation process. A reference to the literature should also be provided here.
Author reply-III: We have included this information in the introduction of the manuscript: “AFm-type phases, monosulfoaluminates, also carbonate in a similar way than that of AFt, leading to calcium carbonates, calcium sulphates and nanocrystalline gibbsite/aluminium hydroxide [18].” (lines 76-78)
[18] Chen, B.; Horgnies, M.; Huet, B.; Morin, V.; Johannes, K.; Kuznik, F. Comparative kinetics study on carbonation of ettringite and meta-ettringite based materials. Cem. Concr. Res. 2020, 137.
R1.4. Chemical formulas are misspelled, eg. "Ca(OH)2" - should be "Ca(OH)2" – remember to use subscripts in the formula.
Author reply: The chemical formulas have been checked along the full manuscript in the word version file and they are ok.
R1.5. Section 2. -I believe that the layout of the subsections must be changed. The PSD and BET studies were described (section 2.7 and 2.8) after their results were reported (section 2.1). I advise to move subsections 2.1 and 2.2 to the end of this section. Rietvield and PDF analyzes are methods of the X-ray diffraction tests interpretation - they should be included in subsections 2.3.1 and 2.3.2, respectively.
Author reply-I: We agree. We have move subsections 2.1 and 2.2 to the end of this section (now they are subsections “2.5. Samples description” and “2.6. Hydration and accelerated carbonation”).
Author reply-II: We also agree. Hence, we have included Rietveld and PDF analyses as a subsection of “Total scattering synchrotron X-ray powder diffraction”. Now: “2.1. Total scattering synchrotron X-ray powder diffraction.; 2.1.1. Rietveld analysis.; 2.1.2. PDF analysis.”
R1.6. l.30:„”- wrong symbol.
Author reply: Thank you for allowing us to clarify this that we (now) understand that it could be misleading. The formula that it was reported in the manuscript was correct: Ca2Si0.972Al0.028O3.986□0.014; “the square” represents oxygen vacancies. This is now clarified in the manuscript: “…with chemical formula Ca2Si0.972Al0.028O3.986□0.014 where the symbol □ stands for oxygen vacancies, …” (line 188)
R1.7. l.137-154. Please better specify the tested samples exact age, because readers may have some doubts about the age of non-carbonated and carbonated samples. What was the storage time for samples in the vacuum sealed bag? Was the storage time of non-carbonated samples extended by 80 days, while the remaining samples were carbonated (this is suggested by Fig. 1)?
Author reply-I: The referee is right and this information was not clear in the manuscript. Now the information has been clarified “To prepare the reference samples, the cylinders were ground by hand in an agate mortar, inserted as powder in an Eppendorf and immediately stored in a vacuum sealed bags by applying about 0.85 bars within a standard food vacuum sealer unit. The samples were stored in the vacuum sealed bag up to the day of data collection (see Figure 1). To prepare the carbonated samples, the cylinders were directly stored in vacuum sealed bags. Prior to introduce the samples in the carbonation chamber, the cylinders were ground by hand in an agate mortar. Accelerated carbonation assay was performed during 80 days at 3% of CO2, at 20ºC and a fixed RH of 65% obtained by a saturated salt solution. The six samples are labelled hereafter: Y-A_hyd, Y-A_carb, C3S_hyd, C3S_carb, C2S_hyd and C2S_carb. It is important to point out that the non-carbonated samples were kept in sealed bags by 80 days, while the remaining samples were carbonated.” (lines 209-217)
Should it be understood that "reference carbonated samples" (l.147) were not carbonated? - This is from the description and is misleading.
Author reply-II: The referee is right. This was a mistake, the sentence is not referred to “reference carbonated samples”, it is referred to “reference samples, not carbonated”. Now in the manuscript: “To prepare the reference samples, the cylinders were ground by hand in an agate mortar, …” (lines 209-210)
R1.8. l.174: “1 to 130° (2θ)” - this is a problematic formulation. The authors do not present full results for this scale. In Figure 2, "Å" was used in place of "°", and other figures used a smaller scale. The descriptions are not consistent.
Author reply-I: Firstly, we would like to note that to be consistent in the x-axis is not possible when we are dealing with two approaches (Rietveld and PDF) from the same raw (total scattering) data as every technique has their needs and ways to display the information. Furthermore, it is also important to clarify that in order to get PDF patterns with very good quality, the recorded Q-range has to be as large as possible. For this reason, for this experiment the use of a short wavelength and high diffracting angles was key. Data were collected from 1 to 130° (2θ) which means Qmax (instrumental) ~25 Å-1. Moreover, as it was stated in the manuscript “PDFgetX3 software [42] was used to obtain the PDF experimental data with Qmax=21 Å-1”. Figure 2 represents full results for this scale with the S(Q) data (not the x-ray diffraction raw data) and this is the reason why we prefer to display the x-axis in Q (Å-1).
Is there a need to present the diffraction patterns for 2θ>60° at all?– they only reduce the readability of fig. 2.
Author reply-II: For the PDF data conversion, we used the total scattering data up to 21 Å-1, we think that is important (at least in the first Figure) to show the full data used for the PDF conversion. On the other hand, if we do not show all the data, some readers could ask why a fraction of the data were left out.
Did the measurement really start from 1°? - it is not visible on the X-ray patterns.
Author reply-III: Yes, the measurements start from 1°. Although for the Rietveld fits, as it was stated in the manuscript, we compute them starting in 2º: “The low angle regions (2-15 º/2q) of the synchrotron total scattering patterns were analysed by the Rietveld method “
R1.9. Section 2.6-2.8. More detailed information on the performed tests should be provided. This is especially important for thermal analysis as entire section 3.3 is devoted to thermal analysis results.
- how were the samples for these tests prepared? - very large mass losses at temperatures below 100 °C, especially for non-carbonated samples, suggest that they may have been damp. In such a case, the presence of free water should be taken into account when making the calculations. In section 2.1. the Authors do not mention that the samples were dried, so I do not consider it correct to combine the total weight loss to 200 °C only with the evaporation of water from “silica gel and the C-S-H phase” (l.428-429 and 460-461).
Author reply-I: For carrying out the thermal analysis studies, the hydration of the samples was not arrested. The procedure, that was described in the Material and Method section, is the following: “the surface was gently dry with paper. To prepare the reference samples, the cylinders were ground by hand in an agate mortar…” Moreover, we underline that most of the calculations were performed in the carbonated samples whose free water should be negligible or very low. However, we cannot exclude the presence of small amounts of FW between RT-200ºC. We have clarified this in the text: “For C3S_carb sample, the first weight loss between RT and ~450ºC, 8.2% is assigned to the water content of the silica gel and the (likely small) fraction of remaining C-S-H gel. Moreover, the presence of a small amount of free water cannot be discarded in this region.” (lines 473-475)
What do the Authors understand by the term "synthetic air" (l.295)? What was the composition of this air? What was the gas flow speed? Why was nitrogen not used?
Author reply-II: We have use a commercial synthetic air with is composed by 80 % of N2 and 20 % of O2. The gas flow speed was 100 ml/ min. We tested that for cement samples, TGA-DTA experiments carry out with synthetic air or nitrogen do not show any difference. We have included this additional information in the material and method section. “The temperature was varied from room temperature (RT) to 1000°C at a heating rate of 10 °C/min and with a gas flow speed of 100 ml/min. Measurements were made in open platinum crucibles under synthetic air flow (80 % N2 and 20 % of O2).” (lines 172-175)
R1.10. All abbreviations should be explained, eg "RT" (first appearance in l.194), "RQPA" (l.220 - referring to the subsection title it should be SRQPA?).
Author reply: The referee is right. We have defined all the abbreviations the first time that they appear in the text. Now: Room temperature (RT); Rietveld quantitative phase analysis (RQPA); Synchrotron X-ray powder diffraction (SXRPD);
R1.11. Fig 3-5. For easier comparison, the same scale should be used for the Y-axis in all X-ray patterns.
Author reply: This is a very cute point and we can only agree. We have modified Figures 3-5 and now all the Y-axes have the same scale.
R1.12. l.338-354 - the description in the text is inconsistent with what can be seen in Figure 8, e.g.:
-l. 341-341: “The most intense interatomic correlation is located at 1.62 Å and it is due to the Si-O bonds.” - in figure 8 for 1.62 Å, the C-O bonds were marked, not Si-O. In Fig. 8b, Ca-O is more intense.
Author reply-I: The referee is right. The sentence was not correct. It has been clarified in the revised version: “One of the most intense interatomic correlation is located at 1.62 Å” (line 380). Moreover, the Si-O interatomic distance was already marked in the Fig. 8b-d at 1.62 Å, the C-O bond was market in the same figures at 1.3 Å.
-343: “…less intense, peaks, at 2.6, 3.0 and 4.1 Å due to O···O…” - in Figure 8, the O···O peak was not marked at all.
Author reply-II: This has been now clarified in the text. “There are three additional, less intense, peaks, at 2.6, 3.0 and 4.1 Å due to O…O, Si…Si and second Si…O interatomic correlations, respectively. The interatomic distance peak of O…O is not labelled in Figure 8 due to its low intensity.” (lines 381-384)
R1.13. l 263: “When this sample is carbonated, several results can be drawn” – sentence incorrect.
Author reply: Thanks. We have rewritten the sentence. Now in the manuscript: “From this study, several results are obtained.” (line 294)
R1.14. l.387: „bassanite aragonite” - lack of comma.
Author reply: The referee is right. We have included this modification.
R1.15. l.29: „[4,5]” – font is too large.
Author reply: We have modified this issue.

Reviewer 2 Report
Attached.

Author Response
Please see the attachment for full letter
-------------
Reviewer #2
In this study, different experimental techniques (including synchrotron-based X-ray diffraction and PDF analysis and TGA) were employed to study the impact of carbonation on the phase formation of several different binders (hydrated alite, belite and ye’elimite). Phase quantification has been performed using all three techniques. There are a number of interesting findings, including the evident formation of amorphous calcium carbonate in the carbonated alite and belite binder but not apparent in the carbonated ye’elimite binder. Overall, this is an interesting paper with high-quality experimental data on different minerals, and hence should be of interest to the readers of Minerals. Hence, the reviewer suggests consideration of publication. Nevertheless, the reviewer does have the following comments/suggestions to help improve the quality of this manuscript.
Author reply: Thank you for the comment. Each point has been addressed below.
R2.1. – Line 25. You mean mineral additives, right?
Author reply: Yes, we have changed in the manuscript “additions” by “mineral additives”
R2.2. Line 27. Please double-check the above figures. The data in the original paper is 2.1-2.3 m3/person/year, which is over 5ton/person/y, if you consider a density of 2.3 ton/m3.
Author reply: Thank you for the correction. We have changed this in the manuscript: “The yearly concrete world production is estimated to be 5 tons/person”
R2.3. Line 33. Check grammar for the first sentence.
Author reply: We have rewritten the sentence. Now in the manuscript: “The carbonation reactions can have a profound impact in:”.
R2.4. Line 35. Why they cannot be discussed here? A brief literature review should be given to identify the research gap.
Author reply: This is exactly the same point raised by the previous reviewer (point R1.2) and it was addressed there. A series of changes have been carried out.
R2.5. Line 59. Do you mean the extent of carbonation?
Author reply: Yes, indeed. We have changed “The extension of the C-S-H carbonation” by “the extent of C-S-H carbonation” (line 61)
R2.6. Line 105. What do you mean by “smooth” here?
Author reply: The referee is right point out that this word was not carefully selected. The sentence was not clear. We have clarified this in the revised version, now: “The analysis of the in situ PDF data showed a continuous calcium decalcification of the C-S-H gel” (lines 123-124)
R2.7. Line 106. Silica-rich?
Author reply: Yes, we mean silica-rich. This has been modified in the main manuscript.
R2.8. Line 143. What type of water? Lime bath?
Author reply: Thank you for the remark. We have specified in the text that we refer to demineralized water.
R2.9. Line 174. Correspond to the wavelength of 0.42318A? or copper K alpha wavelength?
Author reply: Yes, the 1-130º range corresponds to the reported wavelength, 0.42318 Å. This information has been also added in the manuscript: “…in the recorded very wide angular range, 1 to 130° (2θ) (for the employed wavelength, 0.41318 Å).” (line 152)
R2.10. Lines 227-228. It is hard to see this comparison given that they are presented in two figures and the scale of two y-axes is different.
Author reply: This is related to remark R1.11. We have modified Figures 3-5 and now all the y-axes have the same scale.
R2.11. Line 232. What data did you use for Rietveld analysis, I(Q) or S(Q)?
Author reply: For the Rietveld data, the original data, I(Q) were used. This is now clarified in the figure legend of Figures 3-5. “Selected low angle ranges (intensity vs 2q)…”
R2.12. Line 239. Normally C3S hydration does not produce CaCO3 (Unless carbonated).
Author reply: The referee is right. We have completed the sentence in order to clarify the point: “…and the sample does not contain crystalline CaCO3 which proves that the sample was properly prepared and stored.” (lines 267-268)
R2.13. Lines 240-241. Why does this happen? The Rietveld analysis not accurate?
Author reply: As the C-S-H is nanocrystalline/amorphous, this phase was not accounted in the Rietveld quantitative phase analysis that was performed only for crystalline compounds. This issue has also been clarified in the manuscript: However, the main component in this sample is nanocrystalline C-S-H gel being ~67 wt% according to reaction (6) which was not taking into account for the Rietveld quantification due to its nanocrystalline/amorphous nature.” (lines 270-271).
R2.14. Line 256. I don’t think the broad diffusive hump is the background. It is the amorphous phases in your sample.
Author reply: The referee is totally right. The sentence was not clear. We have corrected this in the revised version and we have included a new reference “It should also be noted that the maximum in the background, which arises from the contribution of amorphous phase(s), moved from ~8º (~3.05 Å) for C3S_hyd to ~7º (~3.5 Å) for C3S_carb. The presence of this very large broad hump in carbonated pastes was already reported to be characteristic of amorphous silica gel [52].” (lines 273-277)
[52] Kangni-Foli, E.; Poyet, S.; Le Bescop, P.; Charpentier, T.; Bernachy-Barbé, F.; Dauzères, A.; L’Hôpital, E.; d’Espinose de Lacaillerie, J.-B. Carbonation of model cement pastes: The mineralogical origin of microstructural changes and shrinkage. Cem. Concr. Res. 2021, 144, 106446.
R2.15. Line 258. I guess the beta C2S produced is not pure, so what are the original proportions of the two different C2S phases prior to hydration?
Author reply: The referee is right, there is a lack of information that it is important to understand the hydration of this phase. This information is now included in the main manuscript, in section 2.5. Samples description: “This β-dicalcium silicate sample also contains 8.4(3) wt% of g-dicalcium silicate.” (lines 189-190)
R2.16. Table 4. What is monocarbonate?
Author reply: The referee is right, this phase was not defined. This term is now defined in the footnote of the table (in the revised version): The footnote reads: “*Monocarbonate refers to an AFm-type phase with the following chemical formula: Ca2Al(OH)6[(CO3)0.5.2.5H2O]”. It is also noted that the reference for its structural description was given in Table A1, but we acknowledge that this was not enough.
R2.17. Line 290. This is a strong claim, and how do you know there isn’t any ACC forming? There is still 2-3% difference, and it is possible that to have a few percent of ACC, which is not visible to your XRD and PDF data.
Author reply: The referee is right. We cannot claim that there is no ACC as there is a possibility of having some small amount of ACC that cannot be detected by XRD, DTA and PDF. We have changed the sentence: “ii) all calcium carbonate” by “ii) most of the calcium carbonate” (line 321)
R2.18. Lines 299-300. Why? Could it be (i) small proportions of unreacted Ettringite, (ii) formation of ACC, or (iii) formation of amorphous Al gel, or a mixture of them?
Author reply: There are other possibilities as well. For instance, that amorphous Al(OH)3 gel contains calcium species. We can only speculate here. To address this remark, we have included the following discussion in the revised version “We speculate that this could be due to several reasons including that the amorphous Al(OH)3 gel may contain calcium species and not full decomposition of ettringite according to reaction (3). More research is needed to clarify this disagreement”. (lines 333-335)
R2.19. Line 318. Have you tried to include calcite, monocarbonate and CSH gel?
Author reply: Yes, we tried to include the crystalline phases calcite and monocarbonate and the values went to zero or negative numbers. We also tried to include different structures for the nanocrystalline fraction of the C-S-H gel, although we know that is not an easy task due to its nanocrystalline nature (see references 23, 32, 32 of the manuscript). However, after doing this, the refinement did not improve.
R2.20. Lines 324-329. Needs a bit more explanation of the subtracting process. I guess you calculate the low r region based on the simulated high r region scale factors.
Author reply: The referee is right. This has to be further explained in the text. Now in the manuscript: “This methodology consists on subtracting from the raw patterns, some known contributions, in this case the ones from the crystalline components. For doing this, the parameters for the crystalline phases (scale, unit cell and atomic displacement parameters) that were refined in the 40-60 Å range, are kept fixed in the low PDF region and the difference/subtracted curve corresponds to the contribution of non-crystalline phases (amorphous and nanocrystalline).” (lines 359-363)
R2.21. Line 321. What atom-atom correlations?
Author reply: We agree that this information was missing. We have clarified this point in the manuscript: “Very importantly, the key interatomic correlation at ~3.7 Å (which corresponds to Ca···Ca and Ca···Si) in C3S_hyd and C2S_hyd, highlighted with broken lines in Figure 8”. (line 369)
R2.22. Figure 8. I assume that you have removed all crystalline CaCO3 phases in (b).
Author reply: Yes, as we have further explained in the question/remark R2.20, the fitted parameters for the crystalline CaCO3 phases in the high r-region were kept fixed and the signal of these crystalline phases do not have any contribution in the differential PDF signals shown in Figure 8.
R2.23. Lines 348-350. Which correlation are you referring to? And which two real-space distances.
Author reply-I: The referee is right and the sentence was not clear. We are referring to Si…Si interatomic correlation distance. We have clarified this by replacing: “and this interatomic correlation is more intense.” By “and the band due to the Si…Si interatomic correlation is more intense.”. (line 389)
Author reply-II: Secondly, referring to the two real-space distances, this is now clarified: “… there is scattering in the remaining two real-space distances, at 2.6 and 4.1 Å, but the low intensity of the signals…” (line 391)
It seems that your C3S_carb has more ACC forming than C2S_carb, if you compare Ca-O peak intensity and 4th and 5th ACC peaks. The higher amount of ACC in the former is possibly due to its higher Ca content. I agree that the data is in agreement with the formation of an amorphous silica gel. You could also verify it by comparing the position of the XRD broad diffusive hump with that of silica gel/fume. The intensity of Si-Si is correlated with the amount of Si and the extent of Si polymerization. Higher Si and higher extent of Si polymerization lead to higher Si-Si intensity. A high Ca content (hence possibly high Ca in the Si-rich gel) leads to a lower extent of depolymerization. Both factors could lead to higher Si-Si intensity in C2S_carb than C3S_carb. You could roughly compare the extent of depolymerization by comparing this ratio Si-Si/Si-O (that will need a deconvolution of the peaks.
Author reply-III: According to the referee’s comment, we have measured the integrated intensities of the Si-O and Si-Si peaks for C2S_carb and C3S_carb samples. The intensity of Si-O is 0.89 (for both samples) and the intensity of Si-Si is 0.24 for C2S_carb and 0.11 for C3S_carb. Therefore, the Si-Si/Si-O ratios are 0.27 and 0.12 for C2S_carb and C3S_carb, respectively. However, due to the relatively small differences and the overlap of contributions we prefer not to include this analysis/discussion in the manuscript. However, if the reviewer strongly suggests its inclusion, we will do it.
R2.24. Lines 387-389 and 410-411. You could not eliminate the possibility of forming a small proportion of ACC and/or amorphous Al gel. For example, in the figure above, you have a few percentage weight loss in 550-680 C, which could be related to the decarbonation of ACC.
Author reply: We agree with the referee. We have included this information in the manuscript: “The carbonate measured weight loss, 9.2%, is smaller than the expected one, 11.8%, pointing towards partial presence of calcium within the Al(OH)3 gel as previously reported [57]. Moreover, it is not possible to rule out the presence of a small proportion of ACC and/or amorphous Al(OH)3 gel in the carbonated sample. The thermal analysis trace in Figure 9 (bottom) shows a 2.1% of weight loss in the 550-650ºC range, which could be related to the decarbonation of ACC.” (lines 432-435).
R2.25. Lines 443-445. Not very clear how this is calculated.
Author reply: This has been clarified in the text by replacing “A back of the envelop calculation concerning CaCO3 for C3S_carb would suggest 1.8 moles from C-S-H and 1.2 moles from CH and hence, 60% and 40% of mass losses in the 450-700ºC (Modes II and III) and 700-900ºC (Mode I) regions. The experimental values are 53% and 47% showing a relatively good agreement for such unsophisticated calculation.” By “A back of the envelop calculation concerning CaCO3 for C3S_carb would suggest 1.8 moles from C-S-H and 1.2 moles from CH according to chemical reaction (6). Hence, assuming full carbonation of both phases (C-S-H to give carbonates with weight losses in the 450-700ºC temperature range, Modes II and III, and CH to give crystalline carbonates with weight losses at 700-900ºC, Mode I) the mass losses should be 60% and 40%, respectively. The experimental values are 53% and 47% showing a relatively good agreement for such unsophisticated calculation.” (lines 490-495)
R2.26. Lines 449-450 and line 410. These two sentences do not agree 100% with each other.
Author reply-I: Thank you for highlighting this point. We have changed the value of the temperature in the second sentence/paragraph in order to be consistent. Now in the main manuscript: “The presence of vaterite and aragonite may cause the carbonates to decompose at lower temperatures, and it could define Mode II (700°C<T<780°C). Finally, mode III (550°C<T<700°C) was tentatively to be associated with amorphous calcium carbonate.” (lines 453-456).
Author reply-II: Furthermore, we are aware that we have calculated the weight losses starting at 450ºC instead of 550ºC, but this last temperature was judged to be too high and a significant decomposition seems to be already taken place in our carbonated samples.
R2.27. Lines 440-441. Would ACC transform to crystalline phases during the heating process?.
Author reply: We acknowledge we cannot answer this cute question. In any case, all experimental evidences indicate that the associated weight loss takes place at lower temperatures. This is now clarified in the revised version by adding: “Furthermore, it is currently not known if the ACC phases, in the carbonated pastes, crystallize on heating.” (lines 489-490)
R2.28. Line 459. Any references to support this?
Author reply: The belite hydration could give Ca/Si ratios slightly higher than 1.8. We have recently shown this by PDF methodology and the reference is given [23].
[23] Cuesta, A.; Santacruz, I.; De la Torre, A.G.; Dapiaggi, M.; Zea-Garcia, J.D.; Aranda, M.A.G. Local structure and Ca/Si ratio in C-S-H gels from hydration of blends of tricalcium silicate and silica fume. Cem. Concr. Res. 2021, 143, 106405.
R2.29. Lines 467-468. You have not defined DTA and DTG and label them in the figures, so it would be hard for readers who are not familiar with the technique.
Author reply: DTA was already defined in the material and method section but DTG was not defined.
So, we have corrected this by adding the following statement in the methods section “The derivatives of the weight loss (DTG) have also been calculated and given in the figures.” (line 175). Furthermore and importantly, we have labelled the traces in the TA figures 9-11: (TGA, DTG and DTA).
R2.30. Line 473. Not exactly sure which two effects are you referring to. Mode III/Mode I?
Author reply: We are referring to the ratio between Mode-III (and II) and Mode-I. We have rewritten the sentence to avoid misunderstandings. “The ratio between Modes III-II and Mode I is 75/25.” (line 523).
R2.31. Line 475. Not clear what you are trying to get across here.
Author reply: We noted that there is semiquantitative agreement but not a fully quantitative agreement and more research is needed.
R2.32. Lines 494-495. This is not absolute, right? It is hard to tell from the data if there is a few percent of ACC forming.
Author reply: OK. We have smoothed this sentence by stating in the revised version: “Chiefly, PDF analysis indicated that any amorphous content should be low and no appreciable amounts of amorphous calcium carbonate are formed in these experimental conditions.” (line 545)
R2.33. Lines 497-498. Have you traced their formation as a function of time?
Author reply: No, the data were only collected at the end of the carbonation experiment. This is clearly a way forward (for the research in the near future).
R2.34. Lines 503-504. C-S-H carbonation? Or C-S-H dehydration?
Author reply: C-S-H carbonation.

Reviewer 3 Report
This manuscript studied the carbonation reactions in alite, belite and yéelimite containing pastes by specific microstructure methods. Some interesting results, which is helpful to understand the carbonation of cement phases, could be found. The whole structure is well-organized and the language is suitable for this research. And I believe that this manuscript has an important value for the research on cement hydration and durability from minerals change. Thus, I suggest to accept this manuscript.
Author Response
Please see the attachment for full letter
--------------------------------
Reviewer #3
This manuscript studied the carbonation reactions in alite, belite and yéelimite containing pastes by specific microstructure methods. Some interesting results, which is helpful to understand the carbonation of cement phases, could be found. The whole structure is well-organized and the language is suitable for this research. And I believe that this manuscript has an important value for the research on cement hydration and durability from minerals change. Thus, I suggest to accept this manuscript.
Author reply: Thank you for the comment.

Round 2
Reviewer 1 Report
The manuscript revision was done very carefully. The Authors provided detailed and satisfactory explanations to any comments. However, some minor corrections will still be required.
1. Check the language style - do not use informal phrases, e.g. according to R1.2.: “There is plenty of research in carbonation of building materials and they cannot be discussed here” - this sentence is incorrectly composed…. – the "plenty of" term does not fit to the research article;
2. The Authors affiliations provide in English;
3. The subsection 2.1.1. and 2.1.2. titles should be formatted according to the "Instructions for Author" requirements – non-italic font;
4. Lack of space between subsection 2.4 and 2.5.
Author Response
Reviewer #1
The manuscript revision was done very carefully. The Authors provided detailed and satisfactory explanations to any comments. However, some minor corrections will still be required.
- Check the language style - do not use informal phrases, e.g. according to R1.2.: “There is plenty of research in carbonation of building materials and they cannot be discussed here” - this sentence is incorrectly composed…. – the "plenty of" term does not fit to the research article;
Author reply: We have replaced this sentence, “There is plenty of research in carbonation of building materials…” by “A great research effort in carbonation of building materials has been done…”
- The Authors affiliations provide in English;
Author reply: We acknowledge the good intention of the reviewer. Unfortunately, this advice goes against the University of Malaga guidelines where the affiliation in Spanish is strongly encouraged with a very sound reason. To be homogeneous (and correctly accounted for) in all databases and scientometric studies"
- The subsection 2.1.1. and 2.1.2. titles should be formatted according to the "Instructions for Author" requirements – non-italic font;
Author reply: Thank you. We have changed the format according to the "Instructions for Author" requirements
- Lack of space between subsection 2.4 and 2.5.
Author reply: Thank you. We have included this space.
